# The neutrophil antimicrobial peptide cathelicidin promotes Th17 differentiation

Danielle Minns[1], Katie J. Smith[1], Virginia Alessandrini[1], Gareth Hardisty[1], Lauren Melrose[1], Lucy Jackson-Jones[2], Andrew S. MacDonald [3], Donald J. Davidson [1] & Emily Gwyer Findlay [1✉]

The host defence peptide cathelicidin (LL-37 in humans, mCRAMP in mice) is released from neutrophils by de-granulation, NETosis and necrotic death; it has potent anti-pathogen activity as well as being a broad immunomodulator. Here we report that cathelicidin is a powerful Th17 potentiator which enhances aryl hydrocarbon receptor (AHR) and RORγt expression, in a TGF-β1-dependent manner. In the presence of TGF-β1, cathelicidin enhanced SMAD2/3 and STAT3 phosphorylation, and profoundly suppressed IL-2 and T-bet, directing T cells away from Th1 and into a Th17 phenotype. Strikingly, Th17, but not Th1, cells were protected from apoptosis by cathelicidin. We show that cathelicidin is released by neutrophils in mouse lymph nodes and that cathelicidin-deficient mice display suppressed Th17 responses during inflammation, but not at steady state. We propose that the neutrophil cathelicidin is required for maximal Th17 differentiation, and that this is one method by which early neutrophilia directs subsequent adaptive immune responses.

[1] Centre for Inflammation Research, University of Edinburgh; 47 Little France Crescent, Edinburgh, UK. [2] Division of Biomedical and Life Sciences, Lancaster University, Lancaster, UK. [3] Lydia Becker Institute of Immunology and Inflammation, University of Manchester, Oxford Road, Manchester, UK. ✉email: Emily.findlay@ed.ac.uk

The last 20 years have seen a fundamental shift in our understanding of the roles and capabilities of neutrophils. We are now aware that they can live for much longer than previously thought (up to 5 or 6 days in vivo[1]); that they move into lymphatics, both during inflammation and as part of regular homoeostatic control[2–8]; that they can have reparative as well as pro-inflammatory roles[9–14]; and that they influence outcomes in chronic inflammatory and autoimmune diseases, as well as in infections[15–17]. As a result, we now recognise neutrophils as able to influence and shape adaptive immunity in tissues and in lymph nodes, in both suppressive and activatory manners (reviewed in[18,19]). For example, neutrophils restrain Th2 inflammation during asthma[20], suppress dendritic cell (DC) migration to lymph nodes[21], suppress T cell proliferation in sepsis via Mac-1[22], and limit humoral responses in lymph nodes[23]. In contrast, they can also directly present antigen to T cells via MHC class II[24], cross-present to CD8+ T cells[25], directly stimulate T cell proliferation in response to super-antigen[26], promote type 2 responses in the lung[27], increase DC maturation and co-stimulatory molecule expression[16,28–30] and promote T cell function during influenza infection[31,32].

A particular relationship between neutrophils and the IL-17-producing Th17 subset of CD4+ T cells is now evident. The two cells chemoattract each other to the site of inflammation through production of CCL20/22, IL-17 and IL-8[33], and the production of IL-17A and IL-17F by Th17 cells increases epithelial cell release of G-CSF and CXCL8, increasing neutrophil migration and activation[34]. Following release of extracellular traps, the surviving neutrophil cytoplast structures can indirectly induce Th17 differentiation in the lung during severe asthma[35]. Neutrophils also produce the Th17 differentiation cytokine IL-23[36], their elastase release processes DC-derived CXCL8 into a Th17-promoting form[37], and in vitro they can induce Th17 differentiation directly[38]. However, the mechanisms through which this occurs largely remain unknown.

A major component of neutrophil secondary granules[39,40] and extracellular traps[41] is the short helical antimicrobial host defence peptide (HDP) cathelicidin (human hCAP-18/LL-37, mouse mCRAMP). It can also be produced to a lesser extent by macrophages, mucosal epithelium, eosinophils, adipocytes[42] and mast cells. Release of HDP, such as cathelicidin, is a critical part of the first line innate immune response to infection[43,44] and it is antibacterial, antiviral, antifungal and immunomodulatory[45–47], with potent ability to modulate the local innate and adaptive immune response. Amongst other effects, it can act as a chemoattractant for immune cells[48,49], promote protective inflammatory responses and modulate cell death[50,51], induce wound healing, re-epithelialization and re-endothelialization[52,53], allow the take-up of self-RNA and production of type one interferons by plasmacytoid DC[54,55] and inhibit class switching in B cells[56].

We and others have previously shown cathelicidin to directly and indirectly affect T cell function; it induces DC to prime increased proliferation and pro-inflammatory cytokine production by CD8+ T cells[57], is a chemoattractant for T cells[48], and, in psoriasis, it is recognised directly as an autoantigen by CD4+ T cells[58].

The outcome of neutrophil cathelicidin-CD4+ T cell interaction is largely uncharacterised. In this project we set out to investigate how cathelicidin affected T cell differentiation. We report that cathelicidin is a potent T cell differentiation factor, which induces Th17 and suppresses Th1 differentiation during inflammation, as well as selectively suppressing death of IL-17 producing cells. We show that this is partially dependent on both the aryl hydrocarbon receptor and the presence of TGF-β1 and propose this as a major mechanism by which neutrophils can direct T cell behaviour during inflammatory disease.

## Results

**The host defence peptide cathelicidin induces IL-17A production by CD4+ T cells.** Following on from reports that neutrophils can induce IL-17A production, we hypothesised that cathelicidin specifically affects differentiation of Th17 cells. To examine this we exposed splenic single-cell suspensions from C57BL/6 J mice to Th17-driving conditions. Some wells were exposed to the murine cathelicidin mCRAMP at 2.5 μM, which is a physiologically relevant concentration (with cathelicidin in airway secretions from healthy newborns averaging 2 μM and from those with pulmonary infections up to 6.5 μM[59]). Analysis was performed using the flow cytometry gating strategy in Supplementary Fig. 1.

We noted a consistent increase in the frequency of IL-17A producing CD4+ T cells in culture following cathelicidin exposure under Th17-driving conditions (Fig. 1A, B). The total count of CD4+ cells in culture did not change significantly, from a mean of $33.5 \pm 4.8 \times 10^3$ CD4+ T cells in control samples to $38.1 \pm 5.9 \times 10^3$ cells in cathelicidin-exposed samples. Further, we noted that under non-polarising T cell activation conditions (αCD3/αCD28 stimulation), cathelicidin induced no increase in IL-17A production (Fig. 1C), indicating this peptide is unable to act alone, instead boosting pathways which have already been triggered.

Further investigation with whole splenic cultures revealed that this increase was dependent on both the concentration of cathelicidin present (Fig. 1D) and the time in culture (Fig. 1E); cathelicidin exposure led to a 2.5-fold increase in IL-17A production by day 2 of culture, with further increases seen on day 3. Not only the frequency of IL-17A+ cells but also the intensity of IL-17A expression (Fig. 1F) was increased following cathelicidin exposure, with consequent increased detection of IL-17A by ELISA in cell culture supernatant on day 2 (Fig. 1G).

Th17 cells are characterised by expression of the transcription factor RORγt[60,61]. In keeping with this, cathelicidin also induced strong expression of RORγt in CD4+ T cells following 24 h in culture (Fig. 1H, I). To confirm that our cathelicidin-induced cells are indeed Th17 cells, we co-stained for RORγt and IL-17 and demonstrated that all the IL-17-producing cells induced by cathelicidin were RORγt+ (Fig. 1J).

We confirmed that cathelicidin was enhancing differentiation of these cells directly by isolating splenic CD4+ T cells and exposing them to cathelicidin under Th17-driving conditions (Fig. 1K). IL-17A production by these sorted cells was significantly increased by cathelicidin, with an average 6.2-fold increase in frequency of RORγt+ IL-17A+ cells after 48 h culture and 4-fold increase after 72 h culture (Fig. 1K).

In addition, we assessed phosphorylation of STAT3. STAT3 signalling is required for Th17 but not Th1 differentiation[62]. In isolated CD4+ T cells exposed to cathelicidin for 24 h under Th17-driving conditions, STAT3 phosphorylation was significantly enhanced (Fig. 1L–N).

**Cathelicidin's induction of IL-17 is dependent on TGF-β1 signalling.** Our Th17-driving conditions include αCD3 and αCD28 antibodies, and recombinant IL-6, TGF-β1 and IL-23. To examine whether cathelicidin enhances the signalling of any of these mediators specifically, we assessed its impact on RORγt expression in the presence of each mediator individually. Cathelicidin induced increases in RORγt expression only in the presence of TGF-β1 (Fig. 2A), with no increases seen in cells treated with IL-6 alone, IL-23 alone or IL-6 and IL-23 in combination. Cathelicidin enhanced RORγt in the presence of TGF-β1 alone, but peak expression following cathelicidin exposure occurred in the presence of IL-6 as well as TGF-β1.

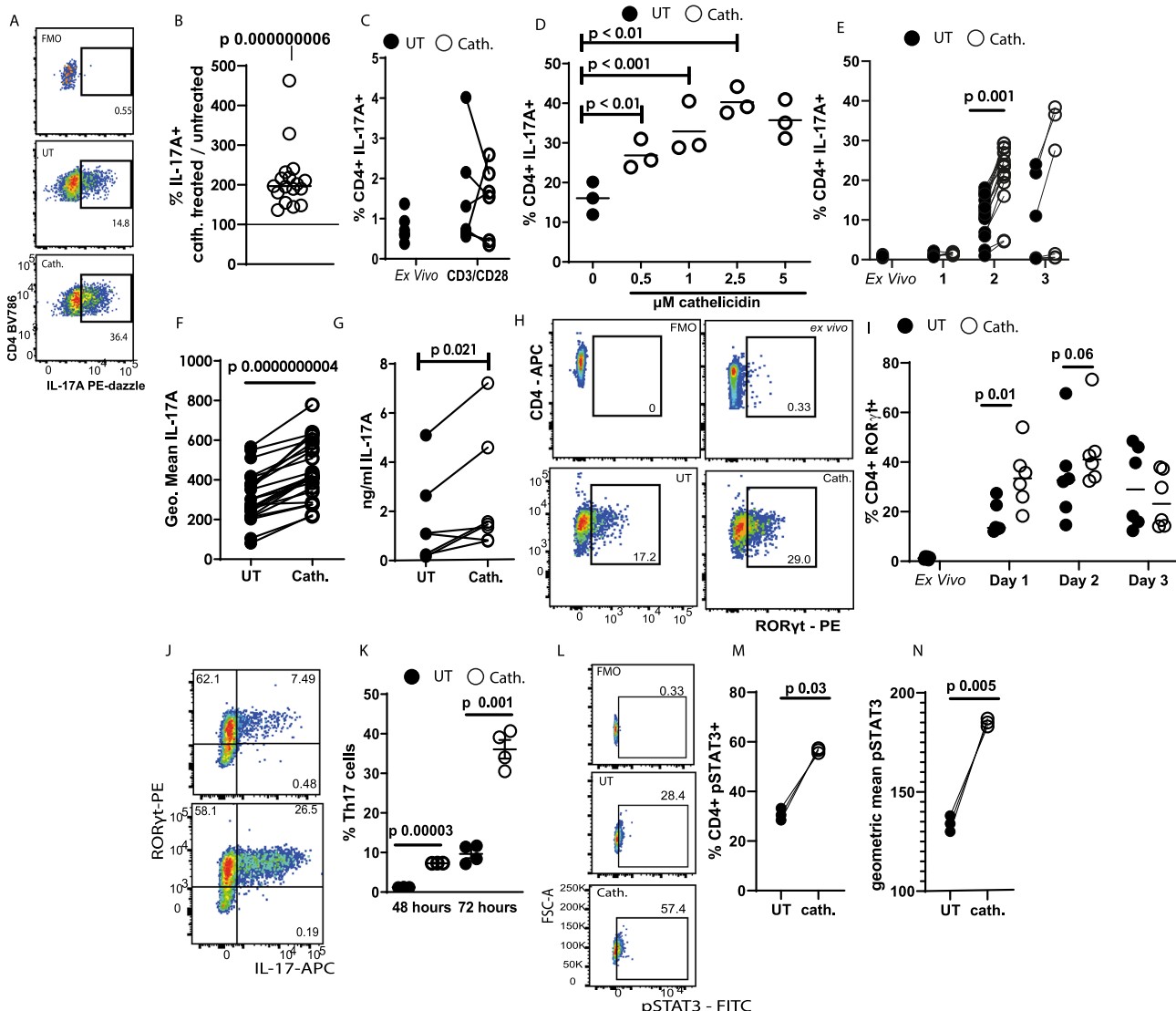

**Fig. 1 The antimicrobial peptide cathelicidin induces IL-17A production by CD4$^+$ T cells.** Splenocytes isolated from C57BL/6 J mice were cultured in Th17-driving conditions for 48 h in the presence or absence of 2.5 µM murine cathelicidin. **A**, **B** Expression of cytokines was determined by intracellular flow cytometry. **C** Splenocytes were cultured with αCD3αCD28 stimulation alone with cathelicidin. **D** Production of IL-17A following increasing doses of cathelicidin (for 48 h) and **E** prodution over increasing days in culture was assessed. **F** Geometric mean of IL-17A expression was also assessed at 48 h. **G** After 72 h of culture supernatant was collected and IL-17A protein quanitified by ELISA. **H** Representative flow cytometry plots showing RORγt expression; **I** RORγt expression was quantified over time in culture. **J** At 48 h in culture all IL-17$^+$ cells were RORγt$^+$. **K** Sorted CD4$^+$ T cells were cultured alone in Th17-driving confitions for 48 or 72 h. **L–N** Following 24 h culture of sorted CD4$^+$ T cells STAT3 phosphorylation was assessed by flow cytometry. Data shown are individual mice used in separate experiments with line at median. Statistical signficance was determined using a one-way ANOVA with Sidak's post-test (**E**, **I**), a two-tailed paired *t*-test with no corrections (**B**, **G**, **F**, **G**, **K**, **M**, **N**) or a one-way ANOVA with a Dunnett's multiple comparison post-test (**D**). *N* values: B - 17, C - 6, D - 3, E - 6, F - 24, G - 7, I - 6, K - 4, M - 3, N - 3. Black symbols represent untreated samples and open symbols represent samples exposed to cathelicidin.

We therefore hypothesised that cathelicidin was enhancing TGF-β1 signalling. It was also possible that the peptide enhanced expression of the TGF-β receptor. Analysis of CD4$^+$ T cells exposed to cathelicidin in Th17-driving conditions revealed that this receptor—and IL-6R and IL-23R—were unaltered (Fig. 2B, C). However, phosphorylation of the Smad2/3 signalling proteins was significantly increased by cathelicidin exposure (Fig. 2D, E). Smad2 and 3 are phosphorylated following TGF-β1 signalling[63] and so this indicated that cathelicidin was indeed enhancing this process.

IL-1β has also been shown to induce Th17 differentiation[64,65]. Cathelicidin induced Th17 differentiation when IL-1β was included in our cultures, but only when TGF-β1 was present,

and this was not enhanced over our normal culture conditions of IL-23, IL-6 and TGF-β1 (Fig. 2F).

Cathelicidin has been shown to use multiple receptors, including P2X7R, CXCR2, GAPDH and FPR2[48,66–70]. It can also enter cells—including plasmacytoid DC, fibroblasts, CHO epithelial cells and bladder carcinoma cells—without a receptor, via lipid-raft dependent endocytosis[55,71]. To determine whether cathelicidin's increase in IL-17A was dependent on a specific receptor, we also used the human cathelicidin LL-37 and the matched D-enantiomer D-LL-37. LL-37 induced the same increase in IL-17A in mouse cells as the murine cathelicidin mCRAMP did; in addition, the D-form peptide D-LL-37 induced a strong increase in IL-17 (UT cells 9.45 ± 0.47%, D-LL-37 16.2 ± 1.01%) (Fig. 2G).

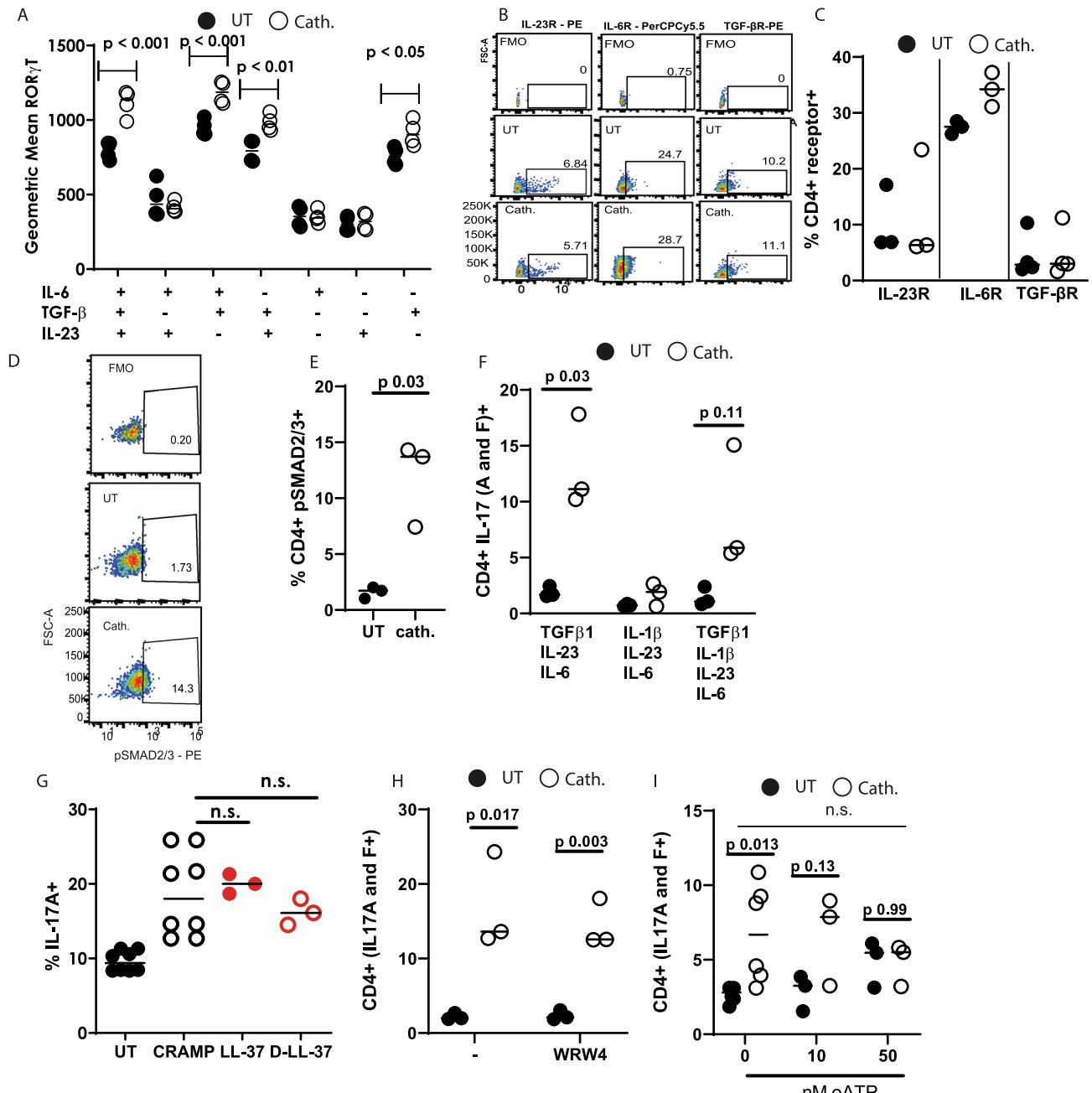

**Fig. 2 Cathelidin-mediated induction of IL-17A is dependent on TGF-β signalling.** Splenocytes isolated from C57BL/6 J mice were cultured in Th17-driving conditions for 48 h in the presence or absence of 2.5 μM cathelicidin. **A** Expression of RORγt following incubation with each component of the Th17 medium was assessed. **B**, **C** Expression of cell surface cytokine receptors was quantified by flow cytometry. **D**, **E** Following 24 h in culture of isolated CD4⁺ T cells, phosphorylation of SMAD2/3 was determined. **F** IL-1β was included in cell cultures and IL-17A production assessed. **G** Different cathelicidin peptides were included in the cultures and **H**, **I** inhibitors of cell surface receptors for cathelicidin. Data shown are individual mice treated separately with line at median. Statistical significance was deetermined using a two-way ANOVA with Sidak's post-test (**A**, **I**), a paired two-tailed *t*-test with no corrections (**E**, **H**, **F**), or a one-way ANOVA with a Tukey's multiple comparison post-test (**G**). *N* values: A - 4, C - 3 for IL-23R and IL-6r, 4 for TGF-bR, E - 3, F - 3, G - 8 for UT and CRAMP, 3 for LL-37 and D-LL-37,.H - 3, I - 6 for 0 nM, 3 for 10 and 50 nM. Black symbols rrepesent untreated samples and open symbols represent samples treated with cathelicidin.

These results suggest that cathelicidin is functioning to modify CD4⁺ T cell phenotype in a manner independent of the peptide binding to a receptor directly.

To examine this further, we performed our Th17 generation assay in the presence of WRW4, an inhibitor of FPR2, and oxidised ATP (oATP), an inhibitor of P2X7 signalling. Including WRW4 in our cultures had no impact on total IL-17 production (Fig. 2H). However, we noted a concentration-dependent suppression of cathelicidin's potentiation of Th17 in the presence of oATP, with 50 nM preventing cathelicidin from enhancing IL-17 production (Fig. 2I). This was not statistically significant owing to an increase in baseline IL-17A production in cells exposed to oATP.

Together, these results reveal that the antimicrobial peptide cathelicidin significantly promotes Th17 differentiation and IL-17A production in a TGF- β1 dependent fashion.

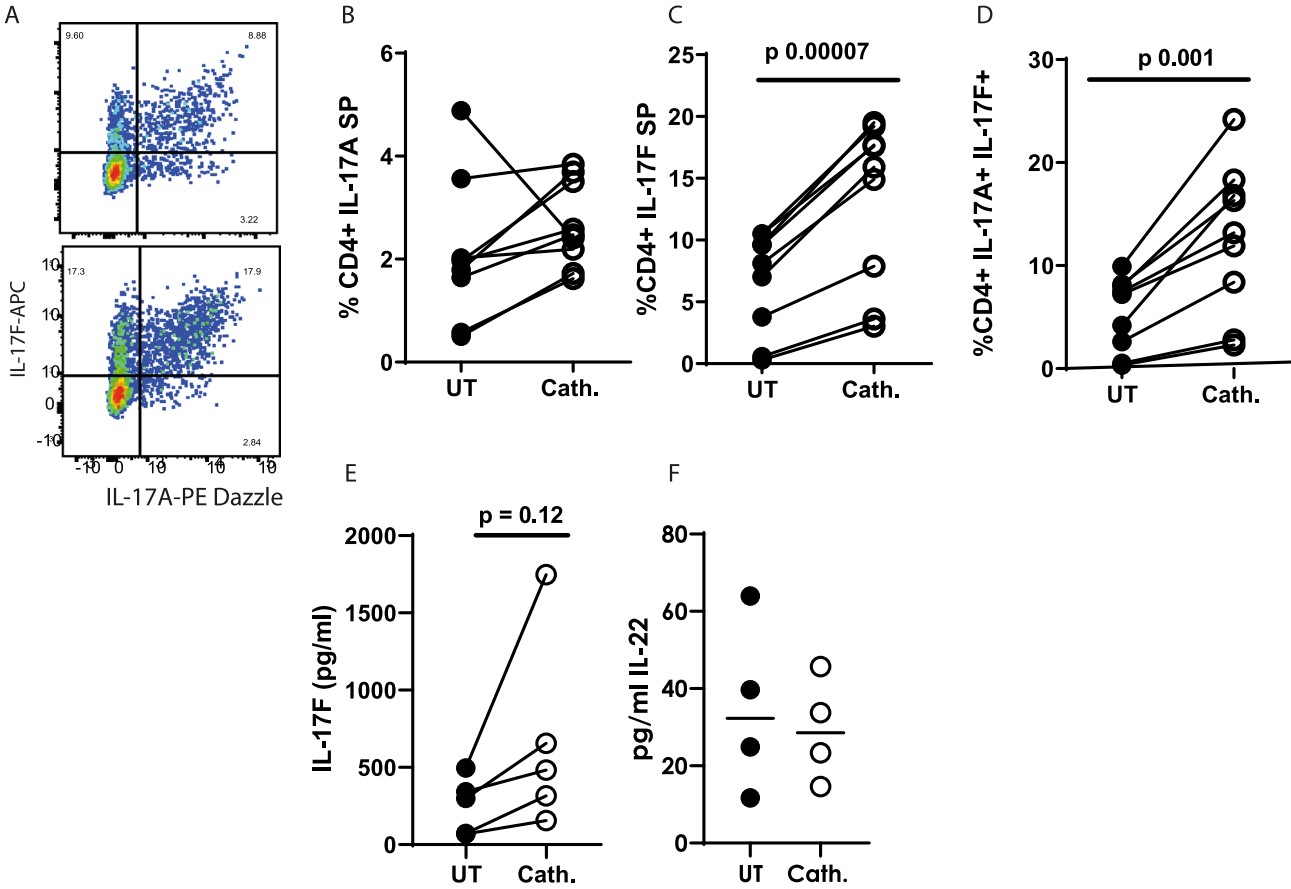

**Fig. 3 Cathelicidin induces IL-17F-producing but not IL-22-producing T cells.** Splenocytes isolated from C57BL/6 J mice were cultured in Th17 conditions for 48 h in the presence or absence of 2.5 μM cathelicidin. **A–D** Expression of IL-17A and F was determined by intracellular flow cytometry. **E** IL-17F and **F** IL-22 production was quantified by ELISA of cell culture supernatant after 48 h in culture. Data shown are individual mice in separate experiments. **B–F** are analysed by paired two-tailed t-tests. SP = single positive. *N* values: B - 9, C - 9, D - 9, E - 5, F - 4. Black symbols represent untreated samples and open symbols are samples treated with cathelicidin.

**Cathelicidin induces IL-17F-producing but not IL-22-producing cells.** Th17 cells produce a number of pro-inflammatory cytokines including IL-17A, IL-17F and IL-22[72–74]. We wondered whether cathelicidin would increase expression of these other cytokines in addition to IL-17A. Analysis of cathelicidin-exposed CD4+ T cells showed, surprisingly, that cathelicidin did not boost IL-17A single positive (SP) cells (Fig. 3A, B). Instead, IL-17F single positive and A+ F+ double positive cells were significantly increased by treatment (Fig. 3C, D). Consequently production of IL-17F in the supernatant, as measured by ELISA, was also increased consistently in every sample, although variability between samples meant this was not statistically significant (Fig. 3E). Interestingly, production of IL-22 was not altered by cathelicidin (Fig. 3F).

**Cathelicidin-mediated enhancement of IL-17A production, but not IL-17F, occurs via the aryl hydrocarbon receptor.** Next, to understand more about the induction of Th17 cells by cathelicidin, we performed gene expression analysis on sorted CD4+ T cells exposed to Th17-driving conditions in the presence or absence of 2.5 μM cathelicidin for 24 hours, using a Nanostring mouse immunology chip. Following cathelicidin exposure, we noted a consistent, significant decrease in the expression of a number of genes which suppress the induction or conversion of cells into the Th17 subset[75–78] including *Socs3*, *Stat1*, *Irf8*, *Bcl3* and *Ikzf4* (Fig. 4A, B).

In keeping with our initial observations, cathelicidin exposure increased expression of the genes coding for IL-17A and the Th17

cytokine IL-21 (Fig. 3A, C). Intriguingly, cathelicidin also led to a large increase in expression of the aryl hydrocarbon receptor (AHR) gene *Ahr* (Fig. 4A, C). Flow cytometric analyses subsequently confirmed an increase in AHR protein following cathelicidin exposure under Th17-driving conditions (Fig. 4D, E), which occurred with increasing time of exposure to cathelicidin and increasing concentration of the peptide present (Fig. 4E, F). The increase in AHR expression by cathelicidin was significant after one day in culture (Fig. 4E), which is earlier than the IL-17 increase was noted.

AHR is a known Th17 differentiation factor[73,75,79] and so we hypothesised that the cathelicidin-mediated increase in IL-17 was a result of its enhanced expression. To test this, we used the specific AHR antagonist CH223191[80]. Use of CH223191 abolished the increase in IL-17A+F+ double-producing cells induced by cathelicidin, indicating that cathelicidin's induction of these Th17 cells is dependent on AHR (Fig. 4G). Interestingly, single IL-17F-producing cells were increased by cathelicidin, as previously shown (Fig. 3C), but this increase was not significantly altered by AHR antagonism (Fig. 4H). These results indicate that there are both AHR-dependent and—independent mechanisms through which cathelicidin promotes Th17 differentiation, with AHR being critical for the enhanced generation of IL-17A producing cells, but not the increase in cells producing only IL-17F.

Finally, we questioned whether cathelicidin-mediated enhancement of AHR was the key factor in Th17 potentiation or whether it

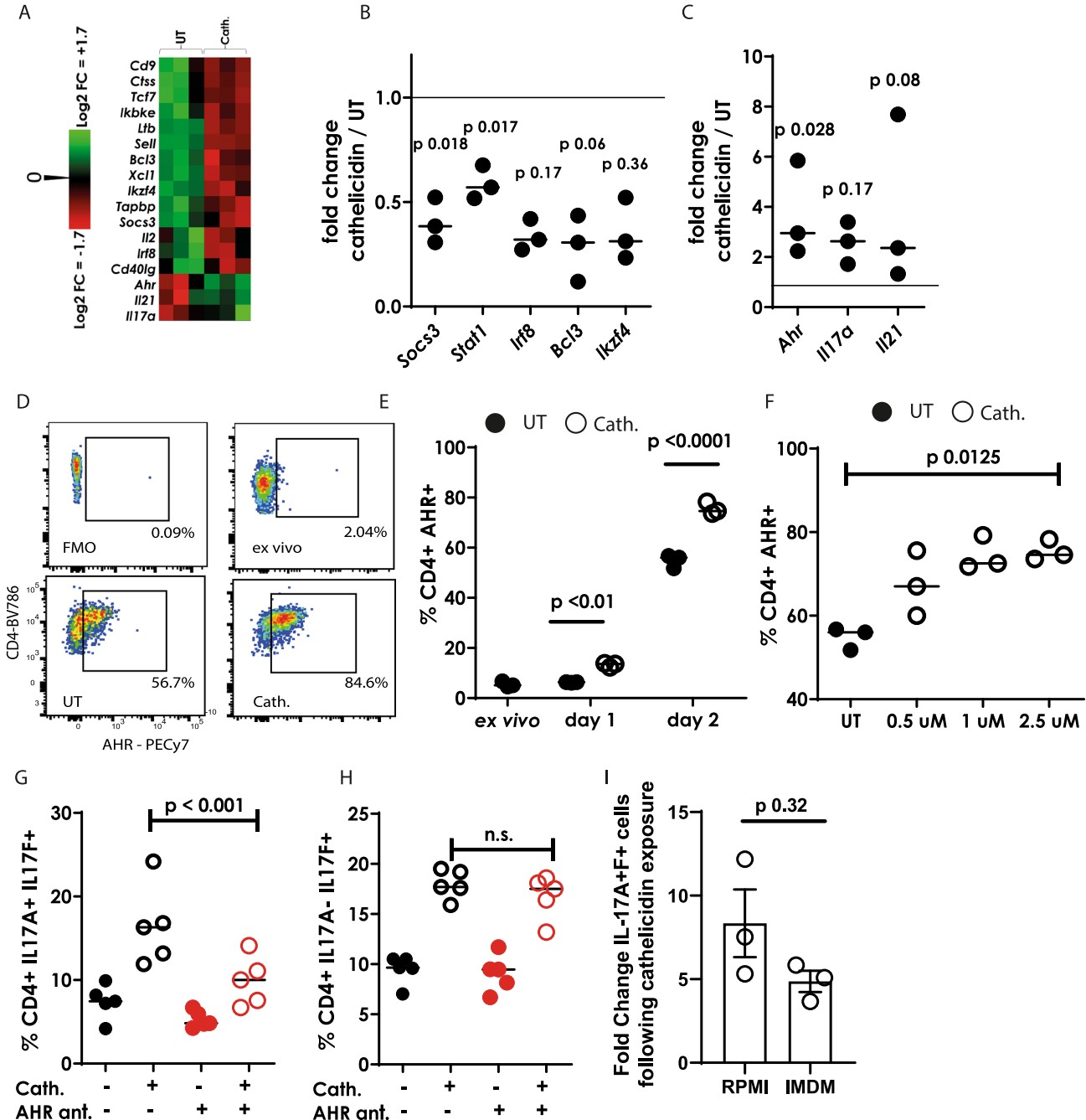

**Fig. 4 Cathelicidin induces Th17 cells via AHR up-regulation.** CD4$^+$ T cells isolated from C57BL/6 J mice were cultured in Th17-driving conditions for 24 h in the presence or absence of 2.5 μM cathelicidin. **A–C** gene expression differences were assessed by a Nanostring mouse immunology chip. **D** Representative flow cytometry plots showing aryl hydrocarbon receptor staining which was assessed over time in culture (**E**) and over increasing concentrations of cathelicidin (**F**). **G, H** IL-17A and F production was assessed by flow cytometry following blockade of AHR with the antagonist CH223191 (ant.). **I** Th17 cultures were repeated using IMDM as well as RPMI medium. Data shown are individual mice with line at medium. Statistical significance was assessed using (**B, C**) a paired two-tailed $t$-test with Bonferroni post-test, **E** repeated measures ANOVA with Sidak's post-test, **F** Kruskal–Wallis test, **G, H** two-way ANOVA with Tukey's post-test and **I** two-tailed $t$-test with no correction. $N$ values: B- 3, C - 3, E - 3, F - 3, G - 5, H - 5, I - 3. The error bars in **I** show standard error of the mean. Black symbols represent untreated samples and open symbols are samples treated with cathelicidin. Red symbols show samples treated with AHR antagonist.

also increased the production of AHR ligands. To answer this, we repeated our Th17 differentiation cultures in IMDM as well as in RPMI. It is known that IMDM contains more natural AHR ligands which induce Th17 differentiation[81]. Cathelicidin enhanced Th17 differentiation equally in RPMI and IMDM (Fig. 4I), indicating that this potentiation occurs even when AHR ligands are in excess. This lends support to our hypothesis that the expression of AHR itself is

the key factor in cathelicidin's enhancement of Th17, meaning that cathelicidin can potentiate AHR signalling and Th17 differentiation at both high and lower ligand levels.

**Cathelicidin suppresses Th1 differentiation in the presence of TGF-β1.** Gene expression analysis also revealed a consistent decrease in Th1-associated genes, including *Il2, Irf8, Stat1*, and

*Xcl1* (Fig. 4A) following 24 hours of cathelicidin exposure in cells differentiating under Th-17 driving conditions. Expression of IL-2 protein was confirmed to be reduced following cathelicidin exposure on day 2 of culture (Fig. 5A), with a striking suppression of IL-2 production by CD4$^+$ T cells, as measured by ELISA. Expression of the Th1-associated transcription factor Tbet (Fig. 5B) and production of the key Th1 cytokine IFN-γ (Fig. 5C, D) were also both strongly suppressed by cathelicidin.

It was very interesting that Tbet was suppressed when RORγt was up-regulated, and we wondered if any double positive (RORγt$^+$ Tbet$^+$ cells) were present in our culture, and whether they expressed IL-17. We found that under Th17-driving conditions almost all the Tbet$^+$ cells also expressed RORγt (Fig. 5E). Tbet single-positive cells were almost abolished in cathelicidin-exposed cultures, and the Tbet$^+$ RORγt$^+$ double positive population was also significantly suppressed. Instead, cathelicidin polarised the cells to RORγt single expressors (Fig. 5E).

Next, we examined the cytokine production by each of these populations, gating on transcription factor expression. We noted that cathelicidin strongly suppressed the tiny populations of Th1 cells (RoRγt- Tbet$^+$ IFNγ$^+$), to confirm our previous findings that cathelicidin suppresses Th1 cell differentiation (Fig. 5F). We also found that cathelicidin enhanced IL-17 production (total IL-17A$^+$ and IL-17F$^+$) by both single expressing RORγt + and double expressing Tbet+ RORγt + cells, although the increase in the latter population was much lower (Fig. 5G, H).

It is well known that Th17-driving conditions do suppress IL-2 production and Th1 commitment[82,83] and we proposed that cathelicidin boosts this suppression further through enhancing TGF-β1 signalling. In keeping with this, we determined that cathelicidin had no impact on Tbet expression in activating non-lineage driving conditions (αCD3, αCD28) but only when TGF-β1 was present (Fig. 5I)—and the presence of TGF-β1 alone was sufficient for cathelicidin to induce this suppression. To confirm this, we analysed production of IFN-γ under Th1-driving and Th2-driving conditions, neither of which include TGF-β1. Cathelicidin did not suppress IFN-γ in these conditions (Fig. 5J). This demonstrates a further enhancement of TGF-β1's action by cathelicidin.

**Cathelicidin promotes survival of Th17 cells but not Th1 cells.** We have shown that cathelicidin promotes increased differentiation of Th17 cells with suppression of Th1 cells in the presence of TGF-β1. We next investigated whether, in addition to modulated differentiation of CD4$^+$ T cells, cathelicidin also led to differential survival or proliferation of particular cell subsets. Cathelicidin has been previously shown to induce death of certain T cell subsets, including T regulatory cells and CD8$^+$ T cells, although CD4$^+$ T effector cells were not affected[84,85]. Our Nanostring gene expression analysis revealed a significant down-regulation of *Fasl* and consistent but not significant down-regulation of *Fas* in sorted CD4$^+$ T cells 24 hours following cathelicidin exposure (Fig. 6A). Subsequently, to determine whether this led to altered rates of death of CD4$^+$ T cells, we measured annexin V staining and uptake of propidium iodide, which together distinguish between live, apoptotic and necrotic cells. Using this method, we noted that cathelicidin significantly suppressed death of CD4$^+$ T cells, with effects seen by the first day of culture (Fig. 6B,C).

To investigate suppression of death by cytokine producing cells in particular, uptake of fixable live dead dyes was assessed (Fig. 6D). Interestingly, cathelicidin exposure suppressed death of IL-17A-producing CD4$^+$ T cells (Fig. 6D, E) but not IFN-γ-producing cells (Fig. 6F), in the same samples, under Th17-

driving conditions. To our knowledge, this is the first demonstration of a neutrophil-released peptide, or of any antimicrobial peptide, increasing survival of T cells. These data raise the possibility that neutrophils may have sophisticated impacts on developing and skewing T cell immunity during inflammatory disease. It was possible that the increased survival of T cells exposed to cathelicidin was related to the altered IL-2 concentrations noted earlier. To test this, we spiked our Th17 cultures with increasing concentrations of recombinant IL-2 and found no alteration in cathelicidin's enhancement of survival (Fig. 6G).

Another possibility was that cathelicidin induces increased proliferation of Th17 cells but not Th1 cells, and that this contributes to their increased frequency in culture. CD4$^+$ T cells labelled with CFSE were observed following exposure to cathelicidin. We noted that there was no difference in the proliferation of RORγt$^+$ Th17 cells at 24 or 48 hours in culture, but at 72 hours differences were noted (Fig. 6H). Surprisingly, cathelicidin induced a suppression in proliferation of RORγt$^+$ cells at this time point, and this was not rescued by addition of recombinant IL-2. The proliferation of Tbet$^+$ cells, interestingly, was not significantly suppressed by cathelicidin (Fig. 6I,J).

**Mice lacking cathelicidin cannot increase IL-17 production in response to inflammation.** Next, we questioned whether our results extended in vivo, and whether T cells differentiate in an altered manner in cathelicidin knockout (*Camp$^{tm1Rig}$*, KO) mice. Firstly we assessed whether KO mice had altered Th17 cell numbers in the steady state. Analysis of ex vivo T cell cytokine production in multiple organs revealed no differences between KO and wild-type C57BL/6JOlaHsd (WT) mice (Fig. 7A,B) – indicating that cathelicidin is not required for development of Th17 cells in the steady state.

We hypothesised that while cathelicidin has no impact on T cell development, it may affect Th17 differentiation during inflammation, when it is released by neutrophils and other cells. To investigate the impact of cathelicidin in the context of inflammation, we used a vaccination model. WT and KO mice were inoculated with heat-killed *Salmonella typhimurium* (HKST) into the top of each hind paw. Seven days later draining popliteal lymph nodes were removed and IL-17A quantified by flow cytometry. KO lymph node CD4$^+$ T cells were unable to produce IL-17A to the same level as WT T cells (Fig. 7C), confirming that cathelicidin promotes Th17 differentiation in vivo as well as in vitro.

This experiment did not tell us if the T cells themselves were inherently different in KO mice or unable to produce IL-17. To test this, we firstly stimulated KO splenic T cells in vitro. In the presence of Th17-driving cytokines, CD4$^+$ T cells from KO mice could produce IL-17 to a level equivalent to wild-type cells (Fig. 7D). Secondly, we performed an adoptive transfer of OT-II CD4$^+$ T cells into WT and KO mice and immunised the mice with ovalbumin in complete Freund's adjuvant. Analysis of donor T cells in draining inguinal lymph nodes and spleens 7 days later showed significantly lower IL-17A production by donor T cells introduced into KO mice (Fig. 7E). In addition, the percentage (Fig. 7F) and number (Fig. 7G) of OT-II cells recovered from KO mice were significantly lower than in WT mice, supporting our earlier in vitro findings that cathelicidin enhances survival of these cells. These two experiments combined led us to conclude that (a) cathelicidin is not absolutely required for Th17 differentiation or for baseline production of IL-17 but (b) in the context of inflammation, cathelicidin potentiates Th17 differentiation and IL-17 production. They also demonstrated that our in vitro observation that cathelicidin is a Th17 differentiation factor was relevant in vivo.

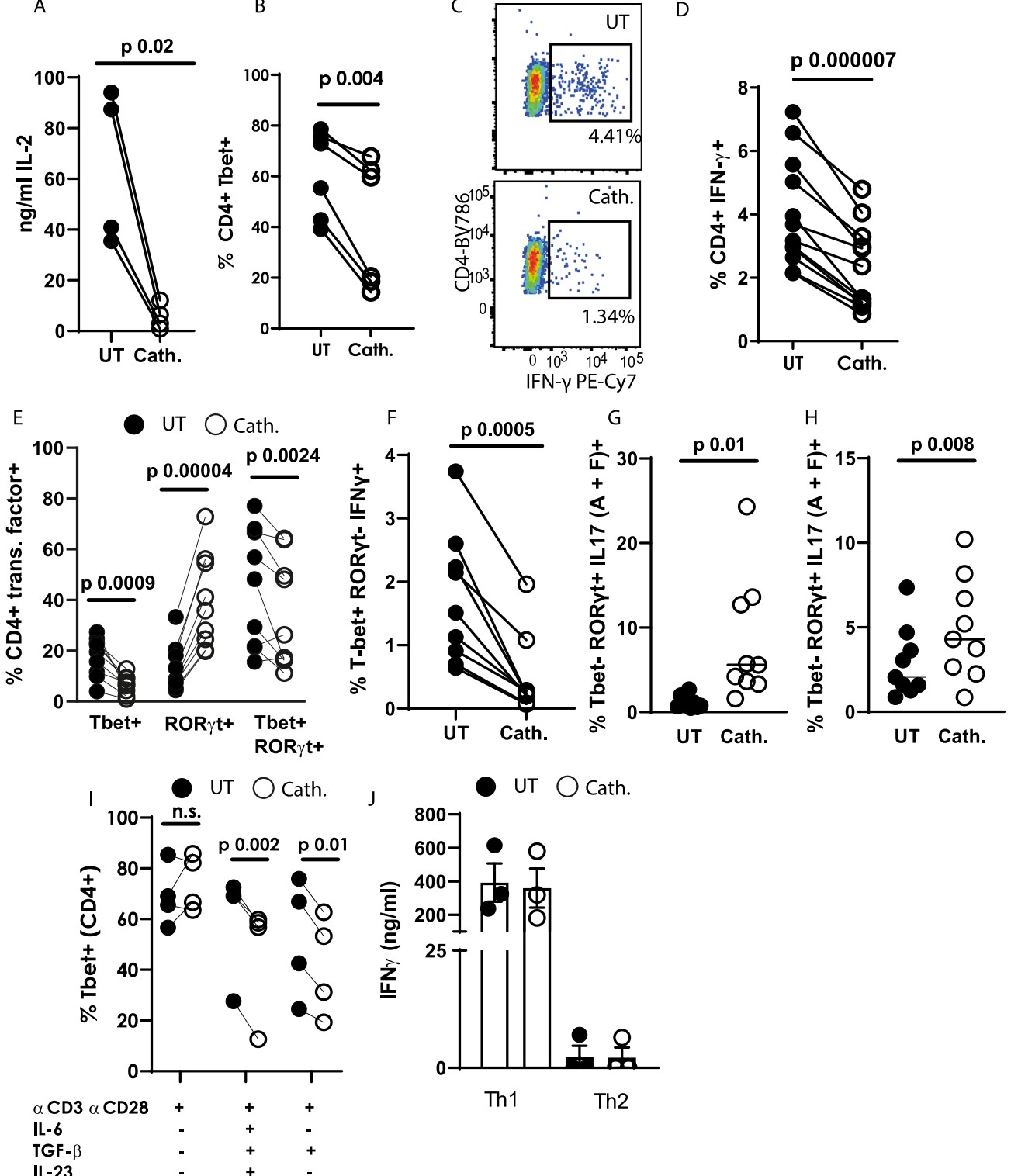

**Fig. 5 Cathelicidin suppresses TH1 differentiation in the presence of TGF-β1.** Splenocytes from C57BL/6 J mice were cultured in Th17-driving conditions for 48 h in the presence or absence of 2.5 µM cathelicidin. **A** production of IL-2 was quantified by ELISA and **B–D** expression of Tbet and IFN-γ were assessed by flow cytometry. **E** Tbet and RORγt co-expression was quantified by flow cytometry and **F–H** the cytokine production by each subset assessed. **I** Tbet expression was determined following incubation with individual cytokines. **J** IFN-γ production was quantified by ELISA after 48 h incubation under Th1-driving or Th2-driving conditions. Data shown are individual mice used in separate experiments. **A**, **B**, **D**, **E**, **F**, **G**, **H** and **I** were analysed by paired two-tailed *t*-tests with no correction, *N* values: A - 4, B - 6, D - 12, E - 9, F - 9, G - 9, H - 9, I - 4, J - 3. The error bars in **J** show standard error of the mean. Black symbols represent untreated samples and open symbols are samples treated with cathelicidin.

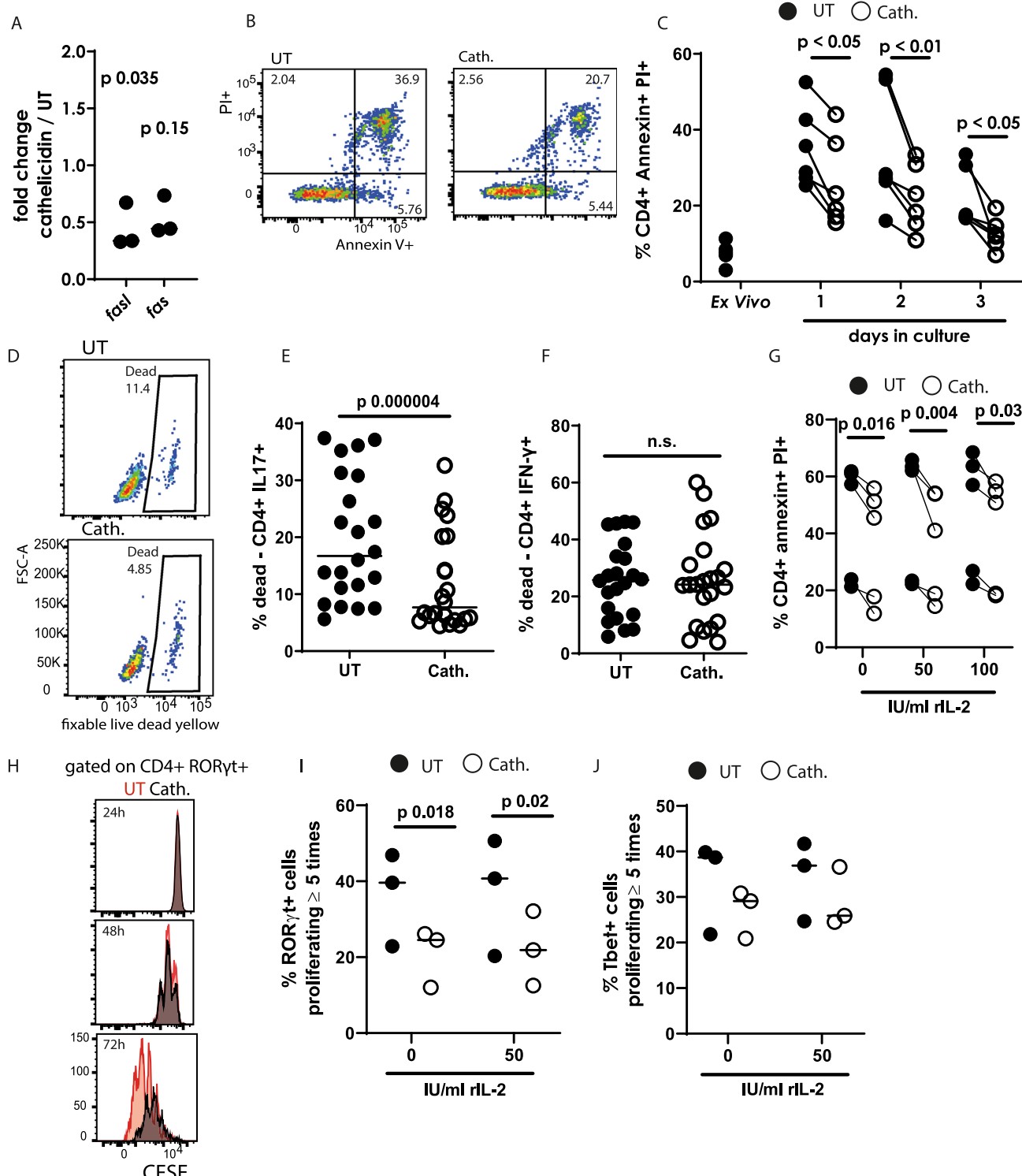

**Fig. 6 Cathelicidin protects Th17 cells but not Th1 cells from death.** CD4 + T cells isolated from C57BL/6 J mice were cultured in Th17-driving conditions for 24 h in the presence or absence of 2.5 μM cathelicidin. **A** Gene expression differences were assessed by Nanostring mouse immunology chip. Cell death was assessed in culture by **B**, **C** annexin V/propidium iodide staining and **D**–**F** binding of a fixable live/dead marker. **G** Annexin/PI staining was repeated at 48 h of culture following inclusion of IL-2 in culture medium. **H**–**J** T cells were labelled with CFSE and proliferation assessed following 24, 48 or 72 h culture —**I** and **J** show 72 h culture analysis. Data shown are individual mice used in separate experiments with line at median. **A**, **E**, **F** and **I** were analysed with paired two-tailed t-tests—**A** and **I** with Bonferroni post correction, **C** and **G** with a two-way ANOVA with Sidak's post-test. N values: A - 3, C - 6, E - 20, F - 19, G - 5, I - 3, J - 3. Black symbols represent untreated samples and open symbols are samples treated with cathelicidin.

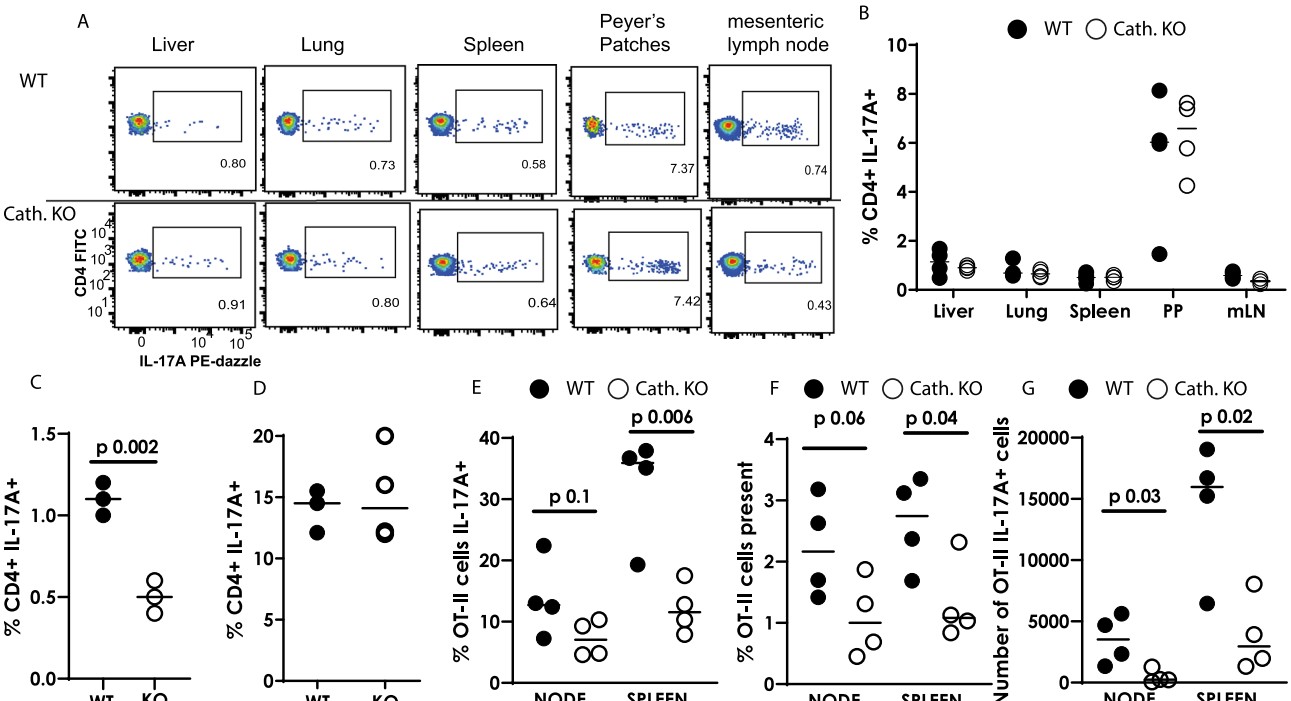

**Fig. 7 Mice lacking cathelicidin cannot increase IL-17A production in response to inflammation.** Wild-type (WT) C57BL/6 J and fully backcrossed cathelicidin knockout (Camp$^{-/-}$, KO) mice were culled and **A**, **B** organs removed and IL-17A production assessed ex vivo by flow cytometry. **C** Mice were inoculated with 12.5 µg/ml heat-killed Salmonella typhimurium and draining lymph nodes removed seven days later—IL-17A was measured by flow cytometry. **D** Splenic T cells from WT or KO mice were stimulated in Th17-driving conditions for 48 h and IL-17A quantified by flow cytometry. **E–G** WT and KO mice were injected intravenously with 5 million OT-II T cells and 24 h given ovalbumin in complete Freund's adjuvant subcutaneously. On day 7 spleens and lymph nodes were removed and donor cell survival and IL-17A production assessed by flow cytometry. Data shown are individual mice with line at median. **C–G** were analysed with two-tailed *t*-tests with no correction. PP—Peyer's Patches; mLN = mesenteric lymph node. *N* values: B - 4, C - 3, D - 4, E - 4, F - 4, G - 4. Black symbols represent wild-type mice and open symbols represent mice lacking cathelicidin.

**Cathelicidin is released in the lymph nodes by neutrophils**. Repeating our in vitro culture experiments with Th17-driving medium, we found that cathelicidin needed to be present in the first 24 hours of culture in order to induce Th17 differentiation (Fig. 8A), with its addition to activated cells leading to no differences in IL-17 production. This suggests that, in vivo, it would only promote Th17 differentiation if present in lymph nodes during the antigen presentation process, and would not have an enhancing effect in the tissue if interacting with T cells which are already activated and differentiated.

We did not know whether or when cathelicidin was produced in the lymph nodes. To determine this, we examined lymph nodes over a time course of HKST injection. Cathelicidin was not present in the steady state lymph node but appeared on day one following inoculation and was strongly present at day 7 (Fig. 8B). Interestingly, we noted the presence of extracellular cathelicidin (arrow heads, Fig. 8B) which had been released from cells.

Co-staining lymph node sections revealed that the vast majority of cathelicidin-expressing cells were neutrophils, with 85% of cathelicidin signal on day 7 being associated with co-staining for the neutrophil marker Ly6G (total cathelicidin+ cells counted over three lymph nodes = 366, of which 55 were Ly6G- and 311 Ly6G + ) (Fig. 8C). We therefore hypothesised that neutrophils were releasing cathelicidin in the lymph nodes and enhancing Th17 differentiation.

Firstly, we confirmed that KO mice did not have an intrinsic defect in neutrophil migration. Following HKST inoculation, neutrophil numbers in the draining lymph node were equivalent between WT and KO mice (Fig. 8D). Next, we examined IL-1β in this system. We did not note a significant change in lymph node IL-1β following HKST inoculation in WT compared to KO mice (Fig. 8E).

We examined whether neutrophil release of cathelicidin could induce Th17 differentiation in vitro. Neutrophil-induced production of IL-17 has previously been demonstrated[86]. Here, we used mouse bone marrow-isolated neutrophils from WT or KO mice, which had been activated by 30 min incubation with fMLF and cytochalasin B, to induce de-granulation. The neutrophils were included in culture wells with WT CD4$^+$ T cells, at a 1:1 T cell: neutrophil ratio and in Th17-driving conditions. WT neutrophils induced IL-17A production by CD4$^+$ T cells (Fig. 8F) but KO neutrophils did not, although the positive control cathelicidin demonstrated that the experiment had worked as normal (Fig. 8G). This data demonstrates that the increase in IL-17A production induced by de-granulating neutrophils is owing to their release of cathelicidin. In addition, AhR expression was significantly lower in T cells co-cultured with KO neutrophils compared to WT neutrophils (Fig. 8H) and blocking cathelicidin from WT neutrophils with an antibody suppressed IL-17A enhancement from sorted CD4$^+$ T cells (Fig. 8I). Finally, we examined responses to an infection model in addition to vaccination. As cathelicidin is directly antimicrobial, it is difficult to decipher its immunomodulatory roles separately to its microbial killing. However, we have previously demonstrated that cathelicidin dramatically enhanced survival in a murine influenza virus infection model, despite only small changes in viral load, suggesting a key immunomodulatory component to protection[87]. We therefore infected WT and cathelicidin KO mice with PR8 influenza and removed the draining mediastinal lymph nodes on day 3 post infection. In agreement with the evidence presented in this paper, IL-17A production in the KO mice was substantially (but not totally) suppressed, when compared to WT mice (Fig. 8J).

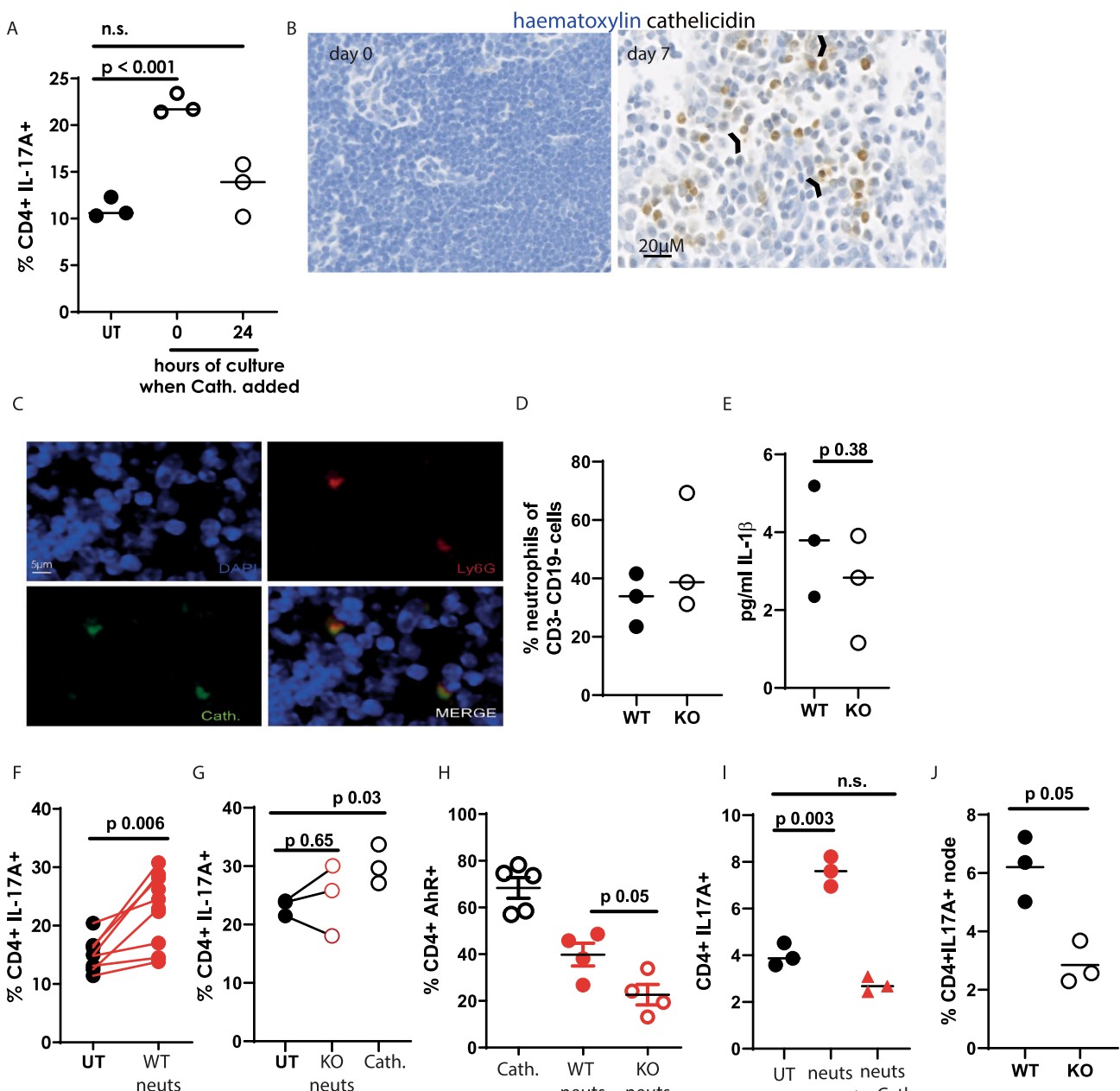

**Fig. 8 Cathelicidin is released in the lymph node by neutrophils. A** IL-17A production was assessed by flow cytometry following cathelicidin being added to mouse Th17-driving cultures on day 0 or day 1. **B**, **C** Wild-type C57BL/6 J mice were inoculated with 12.5 µg heat-killed Salmonella typhimurium (HKST) and draining lymph nodes removed 7 days later. Lymph nodes on day 0 and 7 were assessed for presence of cathelicidin (**B**, arrow heads indicate cathelicidin not associated with cells) and the association of cathelicidin with the neutrophil marker Ly6G (**C**). **D** HKST inoculation of WT and cathelicidin knockout (KO) mice was performed and 24 h later infiltration of neutrophils into the draining popliteal lymph nodes quantified by flow cytometry. **E** IL-1β release was also quantified at 24 h post inoculation. Neutrophils were isolated from the bone marrow of **F** WT and **G** KO mice, primed, and added to Th17-driving cultures of splenic T cells for 48 h before IL-17A production was assessed by flow cytometry and **H** AHR expression was assessed. **I** primed neutrophils from WT mice in the presence or absence of a cathelicidin-blocking antibody were included in Th17 cultures and IL-17A assessed. **J** WT and KO mice were infected intra-nasally with 12pfu PR8 influenza and draining mediastinal lymph nodes removed 72 h later. IL-17A production was quantified by flow cytometry. Data shown are individual mice with line at median, **B** and **C** are representative of 3 experiments. Statistical tests used: **A**, **I** one-way ANOVA with Dunnett's post-test; **E**, **F**, **G**, **H** and **J** paired two-tailed $t$-test with no correction. $N$ values: A - 3, D - 3, E - 3, F - 8, G - 3, H - 5 for cath., 4 for neutrophils, I - 3, J - 3. The error bars in **H** show standard error of the mean. Black symbols—untreated samples, open symbols—cathelicidin treated. Red closed symbols—exposed to wild-type neutrophils, red open symbols—exposed to KO neutrophils. Red triangles—exposed to neutrophils and anti-cathelicidin.

**Cathelicidin induces IL-17A production from human CD4+ T cells**. Finally, to assess whether our observations held true for human cells, we isolated CD4+ T cells from peripheral blood of healthy human donors and incubated them with 2.5 µM human

cathelicidin (LL-37) under Th17-driving conditions. Production of IL-17A was greater in cathelicidin-exposed cells than in controls, measured by flow cytometry on day 8 of culture (Fig. 9A,B) and ELISAs on days 3 and 8 (Fig. 9C). In addition, IL-17F was

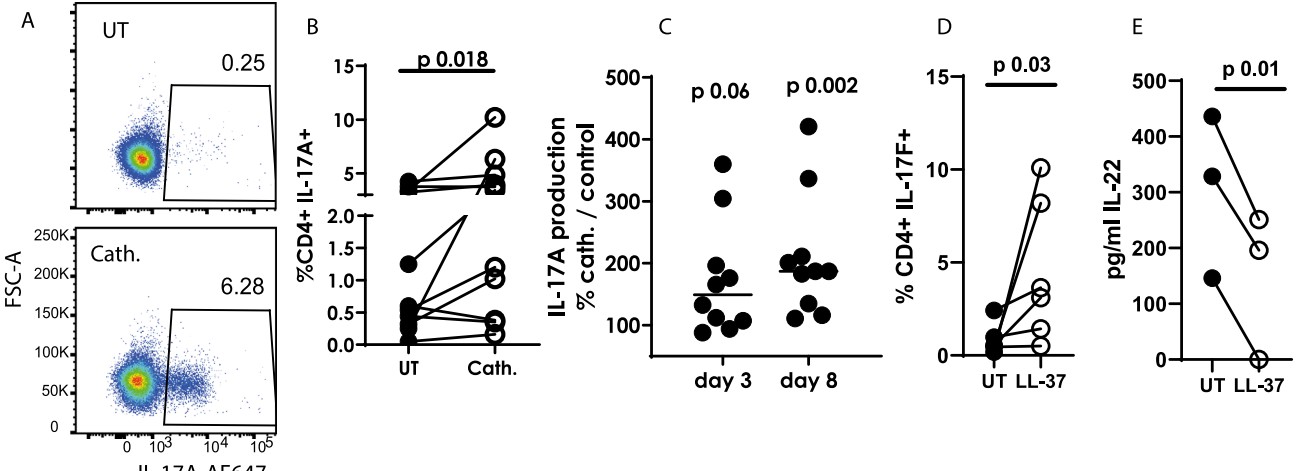

**Fig. 9 Human CD4 + T cells enhance IL-17 production following cathelicidin exposure.** CD4 + T cells were isolated from peripheral blood from healthy human donors and cultured for 8 days in Th17-driving conditions, in the presence or absence of 2.5 μM human cathelicidin (LL-37). IL-17A production was assessed by **A**, **B** flow cytometry on day 8 and **C** ELISA on days 3 and 8 of culture. **D** IL-17F was assessed by flow cytometry and **E** IL-22 by ELISA. Data shown are individual donors with line at mean. **B–E** were analysed by a two-sided Wilcoxon test, on raw data before conversion to %. *N* values: B - 11, C - 11, D - 6, E - 3. Black symbols represent untreated samples, open symbols represent samples exposed to cathelicidin.

also produced at higher levels in response to cathelicidin (Fig. 9D). Interestingly, in contrast, IL-22 production was lower after cathelicidin exposure (Fig. 9E).

## Discussion

We demonstrate that the mouse and human antimicrobial host defence peptide cathelicidin (LL-37/ mCRAMP) induces differentiation of Th17 cells in vitro and in vivo. Specifically, CD4+ T cells exposed to cathelicidin (a) up-regulated AHR, RORγt, IL-17A and IL-17F (b) down-regulated IL-2, IFN-γ and Tbet and (c) survived in greater numbers than their untreated controls. This work extends previous studies showing that neutrophils and Th17 cells amplify each others' responses (reviewed in[18]), providing an explanation for how neutrophils specifically promote one subset of T cells above others.

Cathelicidin exposure led to a 6-fold increase in production of IL-17A in sorted CD4+ T cells (Fig. 1D), and a 2.5-fold increase in production by CD4+ T cells in whole splenic cultures (Fig. 1E). This compares to a 2.5-fold increase in IL-17A production by the aryl hydrocarbon receptor FICZ[75] and a 3-4-fold increase in IL-17A mRNA by the same reagent[73]. These results therefore identify cathelicidin as of equal potency to one of the most important Th17 inducers known.

We show that cathelicidin specifically increases frequency of and cytokine production by those cells which produce IL-17F as well as IL-17A, with seemingly separate pathways promoted in each case. Antagonism of the AHR pathway led only to a suppression in cathelicidin's induction of cells producing IL-17A and not in the induction of IL-17F sole producers (Fig. 3G,H). Previously, triggering of AHR with FICZ led to an increase in IL-17A sole producers and IL-17A+F+ dual producers, but led to a decrease in IL-17F single positive cells[73]. This, along with our data, implies that there may be two pathways promoted by cathelicidin: one of which is AHR-dependent, and which enhances IL-17A+ F+ production; and one which is AHR-independent and which enhances IL-17F production alone. This may also explain the interesting lack of difference in IL-22 production by cathelicidin-treated mouse cells, and its suppression in human cells, as this has also been shown to be driven by AHR ligation[73]. Single-cell transcriptome and signalling pathway analysis of whole splenic cultures treated with cathelicidin will be

required to test this hypothesis, and to extend our understanding of how cathelicidin is enhancing TGF-β1 and AHR signalling.

The up-regulation of RORγt and down-regulation of Tbet induced by cathelicidin was dependent on the presence of TGF-β1 in the differentiation cocktail, and cathelicidin enhanced phosphorylation of SMAD2/3, which mediate TGF-β1 signals[63]. TGF-β1 has been used to differentiate mouse and human Th17 cells in many studies[88,89] and cathelicidin could not induce Th17 cells in its absence, but rather enhanced its effects. Interestingly, cathelicidin did not enhance IL-1 β-induced Th17 differentiation, indicating that the cathelicidin-mediated increase in AHR and IL-17A was a result of enhanced TGF- β1 signalling rather than a more general Th17-boosting phenomenon.

The necessity for TGF-β1 in cathelicidin's induction of Th17 cells extends previous observations of interplay between these two mediators in other cell types[90–92]. Neutrophils can produce TGF-β in lymph nodes and in tissues. Interestingly, lymph node-neutrophil inhibition of B cell activation following inoculation is also TGF-β dependent[23]. Deposition of granule proteins in the lymph node by de-granulating neutrophils has been demonstrated previously[21] and the presence of extracellular cathelicidin in our lymph nodes following HKST inoculum supports our hypothesis that deposition of this peptide is also occurring during inflammation, alongside production of TGF-β from multiple cell types. This allows the enhanced differentiation of Th17 cells during inflammation.

Exposure of CD4+ T cells to cathelicidin under Th17-driving conditions led to an almost complete suppression of IL-2 production, an enhancement of that noted previously as induced by AHR signalling[82]. This observation is intriguing. It would predict that T cells exposed to cathelicidin, in the presence of de-granulating or NETosing neutrophils, will proliferate less, as the available IL-2 is reduced. Further, neutrophil-induced suppression of T cell proliferation is well documented[22,93]. Indeed, we demonstrate that cathelicidin inhibited proliferation of Th17 cells in culture and therefore suggest that the neutrophils' release of cathelicidin, and subsequent suppression of IL-2 production, is one mechanism behind this phenomenon.

The increased survival specifically of Th17 cells following cathelicidin exposure is interesting. Neutrophil-boosted survival of any T cell subset has never previously been described. Neutrophils can release BAFF and APRIL in the lymph nodes and

thus protect B cells against apoptosis[94–96], and can also enhance NK cell survival[97], meaning our demonstration of T cell survival fits within the context of the wider lymphocyte population.

Th17 cells are associated with a number of inflammatory and autoimmune conditions, including psoriasis, rheumatoid arthritis, and type 1 diabetes (reviewed in ref. [98]). Our work supports previous data showing that cathelicidin is recognised by some CD4[+] T cells as an autoantigen during psoriasis, and those T cells which recognise cathelicidin produce more IL-17[58]. The phenotype of the T cells resulting from cathelicidin exposure–proliferating less, but surviving longer, and producing more pro-inflammatory cytokines – suggests that they would be deleterious long-term, and therefore that neutrophils present during these conditions could be responsible for increased disease onset and severity. In this regard, however, the specific increase in IL-17F producing cells by cathelicidin is interesting, as this cytokine has been correlated with non-pathogenicity of cells[99,100], and the spinal-cord infiltrating Th17 cells in EAE are mostly solely producing IL-17A[100]. Future work understanding the role of neutrophil-released cathelicidin in altering the pathogenicity of Th17 cells, and in the specific induction of IL-17F, will be of great interest.

Whether cathelicidin enters T cells through a specific receptor is unclear. We demonstrate that the D-enantiomer of human cathelicidin induces Th17 differentiation, strongly suggesting the peptide can enter T cells in a receptor-independent fashion, as it can do to other immune cells. However, our data showed a suppression of Th17 differentiation in the presence of oATP, an inhibitor of the receptor P2X7. This was not statistically significant, and interpretation is complicated by a) significantly enhanced cell death in T cells exposed to oATP, b) an increase in baseline IL-17 production in T cells exposed to oATP but not to cathelicidin, and c) the fact that oATP is not only a P2X7 inhibitor but also attenuates inflammation by P2X7-independent mechanisms[101]. We infer from these data that there may be an indirect mechanism involving P2X7 in cathelicidin-mediated Th17 differentiation, but unravelling this mechanism will require knockout mice or more precise inhibitors than oATP.

Overall, our study identifies the antimicrobial peptide cathelicidin as a key Th17 differentiation factor and defines one mechanism by which neutrophils can alter developing T cell responses in a sophisticated and specific manner.

## Methods

**Experimental design**. All experiments were performed three times on individual mice or donor cell cultures and then power calculations performed to determine necessary group sizes. Both male and female mice and human donors were used in experiments and no differences were noted in responses between the sexes. Numbers of individuals and of experiments are listed in each figure legend.

Mice were housed together for in vivo experiments to avoid cage-specific effects.

**Mice**. Wild-type C57Bl/6JOlaHsd and mCRAMP knockout (Camp[tm1Rig], KO) mice were bred and housed in individually ventilated cages, under specific pathogen-free conditions. Male and female mice between 6-12 weeks of age were used. KO mice were backcrossed onto wildtypes for 10 generations before use. Mice were kept in environmental conditions in line with the ASPA Code of Practice for the UK – temperatures between 19 and 24ºC, humidity between 45 and 65%, and 12 h of light then 12 h of dark.

All animal experiments were performed by fully trained personnel in accordance with Home Office UK project licences PAF438439 and 70/8884, under the Animal (Scientific Procedures) Act 1986. All relevant ethical regulations for animal research were fully complied with. The experimental programme of work was described in project licence PAF438439 and was approved as such. Each experimental protocol was overseen by the in-house University of Edinburgh veterinary team.

**Healthy human donors**. Peripheral venous blood was collected from healthy adult volunteers under ethical agreement code AMREC 15-HV-013. All ethical regulations from the University of Edinburgh were adhered to and informed written consent was obtained from every donor. Experimental programmes were overseen by the University of Edinburgh Centre for Inflammation Research Blood Resource Management Committee. Up to 160 ml of blood was collected at one time into sodium citrate and was then processed immediately. Ficoll Paque Plus (GE Healthcare #GE17-1440-02) was used to isolate mononuclear cells by mixing with freshly isolated blood (1:1 diluted in PBS) and spinning in LeucoSep tubes (Grenier #227289) for 15 min at 1000 g, with brake at 0. CD4[+] T cells were then isolated from the cell layer using the EasySep[TM] human CD4[+] T cell isolation kit (StemCell Technologies #17952), according to manufacturers' instructions.

**Human Th-17 cultures**. Human CD4[+] T cells were plated at 1×10[5] cells/well in a round-bottom 96-well plate in complete medium (RPMI, 10% foetal calf serum, 10 units/ml penicillin, 10 μg/ml streptomycin and 2 mM L-glutamine, all supplied by Gibco, ThermoFisher UK). Cytokines added were: 10 ng/ml TGF-β1 (Biolegend #580702), 100 ng/ml IL-6 (Biolegend #570802), 30 ng/ml IL-1β (Biolegend #479402), 30 ng/ml IL-23 (Biolegend #574102) and 2.5 μl/ml anti-CD3/23/2 ImmunoCult T cell activator (StemCell Technologies #10970). Medium was changed on day 3 and samples were assessed by flow cytometry and ELISA on day 8.

**Peptides**. Synthetic mCRAMP (GLLRKGGEKIGEKLKKIGQKIKNFFQKLVPQ PEQ) and LL-37 (LLGDFFRKSKEKIGKEFKRIVQRIKDFLRNLVPRTES) were custom synthesised by Almac (Penicuik, Scotland) using Fmoc solid phase synthesis and reverse phase HPLC purification. Peptide identity was confirmed by electrospray mass spectrometry. Purity (>95% area) was determined by RP-HPLC and net peptide content determined by amino acid analysis. D-enantiomer LL-37 was a kind gift from Professor Peter Barlow, Edinburgh Napier University. Lyophilised peptides were reconstituted in endotoxin free water at 5 mg/ml. Reconstituted peptides were tested for endotoxin contamination using a Limulus Amebocyte Lysate Chromogenic Endotoxin Quantitation Kit (Thermo Scientific, UK #88282).

**Murine tissue and single-cell preparations**. Mice were culled by rising concentrations of $CO_2$, followed by cervical dislocation. Single-cell preparations of spleens, lymph nodes and Peyer's patches were achieved by mashing the tissues through a 100 μM strainer and washing with PBS. One lung lobe was minced and enzymatically digested with 1 mg/ml collagenase VIII (Sigma-Aldrich, #C2139) for 20 mins at 37 °C, with shaking before the cell suspension was passed through a 100 μM filter. To isolate liver leucocytes, one lobe was passed through a 100 μM strainer and leucocytes isolated by Percoll separation[102]. To do this, cell pellets from straining were re-suspended in 32% Percoll, 3% 10xPBS, and 65% Hanks balanced salt solution. The cells were spun at 600 g for 10 min. From this, floating hepatocytes were removed and leucocytes collected from the pellet.

In all cases red blood cells were lysed using RBC Lysis Buffer, as per the manufacturer's instructions (BD Biosciences, #555899).

**Isolation of bone marrow-derived neutrophils**. Femurs were removed and marrow flushed out with complete medium (RPMI, 10% foetal calf serum, 10 units/ml penicillin, 10 μg/ml streptomycin and 2 mM L-glutamine, all Gibco, ThermoFisher, UK). Single-cell suspensions were prepared by passing the cells through a 19 G needle. Neutrophils were isolated using the EasySep[TM] Mouse Neutrophil Enrichment Kit (StemCell Technologies, #19762), as per the manufacturer's guidelines. Neutrophils were activated by 30 min incubation with 100 nM fMLF (N-formylmethionine-leucyl-phenylalanine, Sigma-Aldrich #F3506) and 10 μM cytochalasin B (Sigma-Aldrich #C2743) before being thoroughly washed twice.

**In vitro T helper cell subset differentiation**. Whole splenocytes were prepared as above. CD4[+] T cells were isolated from whole splenocytes by FACS sorting on a BD Biosciences Aria sorter using a 70 μM nozzle or using the EasySep mouse CD4[+] T cell isolation kit (StemCell Technologies, Cat #19852). Purity achieved with both methods was always over 95%.

200,000 cells were plated per well of round-bottom 96-well plates in complete medium (RPMI, 10% foetal calf serum, 10 units/ml penicillin, 10 μg/ml streptomycin and 2 mM L-glutamine, all supplied by Gibco, ThermoFisher UK), with the correct combination of cytokines and neutralising antibodies as follows. Th1 cells were differentiated for 4 days in the presence of plate-bound αCD3 (5 μg/ml; Biolegend, #100339), rIL-12 (25 ng/ml; Biolegend, #575402), rIL-18 (25 ng/ml; Gibco, #PMC0184) and rIL-2 (10 U/ml; Biolegend, #575402), with or without 2.5 μM mCRAMP. Th2 cells were cultured for 4 days with plate-bound αCD3 (5 μg/ml; Biolegend, #100339), rIL-4 (4 ng/ml; Peprotech, #214-14), rIL-2 (40 U/ml; Biolegend, #575402), αIL-12 (5 μg/ml; Biolegend, #505307) and αIFNγ (5 ug/ml; Biolegend, #505812), with or without 2.5 μM mCRAMP. Th17 cells were differentiated in the presence of plate-bound αCD3 (5 μg/ml; Biolegend, #100339), rIL-6 (20 ng/ml; Biolegend, #575706), rIL-23 (20 ng/ml; Biolegend, #589006) and rTGFβ (3 ng/ml; Biolegend, #580706), with or without 2.5 μM mCRAMP for 1 to 3 days. αCD28 (1 μg/ml; Biolegend, #102115) was added to cultures of pure CD4[+] T cells. In some experiments neutrophils, isolated as above, were incubated in CD4[+] T cell cultures at a 1:1 neutrophil: T cell ratio. In other experiments

recombinant IL-2 (Biolegend, #575402) or recombinant IL-1β (RnD Systems, #401-ML) were included in cultures.

**Flow cytometry**. Cells were stained for surface markers for 30 mins at 4 °C, protected from light. Intracellular cytokines were assessed by incubating cells for 4 h at 37°C with Cell Stimulation Cocktail containing protein transport inhibitors (eBioscience, #00-4970-03). Cells were fixed, permeabilised and stained for cytokines using BD Cytofix/Cytoperm (BD Biosciences, #554722), as per the manufacturer's guidelines. Cells were fixed, permeabilised and stained for transcription factors using the True-Nuclear Transcription Factor Buffer Set, as per the manufacturer's instructions (Biolegend, #424401). Cell viability was assessed by flow cytometry using the Annexin-V-FLUOS Staining Kit, as per the manufacturer's guidelines (Roche, #11 858 777 001) or fixable live/dead yellow (ThermoFisher #L34959). To assess proliferation, cells were stained with carboxyfluorescein succinimidyl ester at 1:2000 dilution for 10 min (Invitrogen, #C34554) before culture. Proliferation analysis by dye dilution was performed by flow cytometry. Samples were collected using an LSRFortessa cytometer(BD Biosciences) and Diva software version 9, and analysed using FlowJo software version 10 (BD Biosciences).

Flow cytometric analysis of phosphorylated antigens was performed by first staining surface markers for 30 mins at 4 °C, before adding 1 ml of phosflow lyse/fix buffer (BD Biosciences, Cat #558049) for 60 min at room temperature. Cells were then centrifuged and resuspended in 1 ml ice-cold methanol for 30 min on ice before being washed with PBS. Antibodies in PBS were used to stain intracellular phosphorylated antigens for 45 min before being washed and data collected immediately.

**Antibodies (mouse)**. CD4 (clone GK1.5, Biolegend, #100453, dilution 1:200); CD8 (53-6.7, BD Biosciences, #563786, 1:200); IFNγ (XMG1.2, Biolegend, 505825, 1:100); IL-17A (TC11-18H10.1, Biolegend, #506938, 1:100); IL-22 (POLY5164, Biolegend, #516411, 1:100); RORγT (B2D, eBiosciences, #12-6981-80, 1:100); TBET (4B10, Biolegend, #644805, 1:100); AHR (4MEJJ, eBiosciences, #25-5925-80, 1:100); pSTAT3 (LUVNKLA, eBiosciences, #11-903-42, 1:100); pSMAD2/3 (o72-670, BD Biosciences, #562586, 1:100)

**Antibodies (human)**. CD4 (clone OKT4, eBioscience #25-0048-42, 1:200), IL-17A (eBio64DEC17, eBioscience #12-7179-42, 1:100), IL-17F (Poly5166, Biolegend #516604, 1:150).

**Nanostring**. Mouse Th17 cultures were set up as previously described. DAPI⁻γδ⁻CD4⁺ T cells were sorted using a BD FACSAria™ II (BD Biosciences) on day 1. RNA was extracted immediately using the Qiagen RNeasy Mini Kit (Qiagen, #74104), as per the manufacturer's guidelines. Multiplex gene expression analysis (Mouse Immunology Panel) was performed by HTPU Microarray Services, University of Edinburgh. Data analysis was performed using nSolver 4.0 and nCounter Advanced Analysis software. Genes with a log2 fold change of 1 or more and a p value of 0.05 or less, following correction for multiple comparisons, were deemed of interest and statistically significant.

**ELISA**. Concentrations of mouse IL-17A (Biolegend, #432504), IFNγ (Biolegend, #430804), IL-2 (Biolegend, #431004), IL-17F (Biolegend, #436107) and IL-22 (Biolegend, #436304), and of human IL-17A (Invitrogen #BMS2017) and IL-22 (ThermoFisher, #BMS2047) were determined in cell culture supernatants by ELISA, as per the manufacturer's guidelines.

**Immunohistochemistry**. Lymph nodes were fixed in 10% neutral buffered formalin and embedded in paraffin. 10 μm sections were deparaffinized and antigens retrieved by microwaving in tri-sodium citrate buffer, pH 6 for 10 min, or by 10 min incubation with 5 mg/ml proteinase K (ThermoFisher UK #AM2548). Slides were blocked (Avidin/Biotin Blocking Kit, Vector Laboratories, #SP-2001) and incubated overnight with rabbit anti-mCRAMP (1/250; Innovagen, #PA-CRPL-100). Slides were then incubated with an anti-rabbit HRP-conjugated secondary antibody (1/500; Dako, #P0217) or AF488-conjugated antibody (1:500, ThermoFisher UK #A11034). Some sections were developed with diaminobenzidine (ImmPACT DAB Peroxidase (HRP) Substrate, Vector Laboratories, #SK-4105) and others were also incubated with an AF647 anti-neutrophil antibody (1:50, Biolegend UK #127609). Slides were scanned on a ZEISS Axio Scan.Z1 slide scanner.

**Heat-killed Salmonella model**. Heat-killed salmonella was a kind gift from Professor Andrew Macdonald, MCCIR, University of Manchester. 12.5 μg heat-killed *Salmonella typhimurium* in 50 μl PBS was injected subcutaneously in the top of each hind paw. Mice were monitored and the draining popliteal lymph nodes removed after 1, 3 or 7 days. Nodes were either fixed in 10% neutral buffered formalin and embedded in paraffin for sectioning, or single-cell suspensions were prepared as described previously for flow cytometric analysis.

**Influenza infection model**. Murine influenza strain PR8, a kind gift from Dr Ananda Mirchandani, University of Edinburgh, was instilled intra-nasally into mice—12p.f.u. in 40 μl inoculations. Mice were monitored daily and the draining

mediastinal lymph nodes removed 72 h later. Flow cytometry on node single-cell suspension was performed as previously described.

**Statistics**. All data shown are expressed as individual data points with line at median. Analysis was performed with GraphPad Prism software. Multiple groups were compared by one- or two-way analysis of variance tests with either Bonferroni or Dunnett post-tests. Two groups were compared with two-way paired Student's *t*-tests. A minimum of three mice was used for in vitro experiments, in individually performed experiments. Details of sample sizes are included in all figure legends.

**Reporting summary**. Further information on research design is available in the Nature Research Reporting Summary linked to this article.

## Data avaiibility

The datasets generated during the current study are available from the corresponding author (EGF) upon request. The authors declare that all data supporting the findings of this study are available within the paper.

The Nanostring data set has been deposited and can be found at the Gene Expression Omnibus repository with accession code GSE160757.

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

## Acknowledgements

We thank the Queen's Medical Research Institute flow cytometry facility (Shonna Johnston, Mari George and Will Ramsey) and the Centre for Cancer Research, University of Edinburgh Nanostring team (Alison Munro) for help and advice, and Dr Robert Gray and Professor Julia Dorin for helpful discussions.

## Author contributions

Conceptualisation: DM, DJD, EGF; Funding acquisition: EGF, DJD; Investigation: DM, KJS, VA, LM, GH, LJJ, EGF; Methodology: AMcD,EGF; Project administration: EGF; Resources: AMcD, DJD, EGF; Supervision: DJD, EGF; Writing – original draft: DM, EGF; Writing – review and editing: DM, EGF, DJD.

## Funding

This work was funded by a Royal Society Dorothy Hodgkin Fellowship (DH150175), a Royal Society Fellows' Enhancement Award (RGF\EA\180049), a Tenovus Scotland project grant (E17/01) and a Carnegie Scotland Research Incentive award (RIG008679), all to EGF, and by a Medical Research Council Senior Fellowship (G1002046) to DJD.

## Competing interests

The authors declare no competing interests.
