## [Peer Review File · Nature Communications]

REVIEWER COMMENTS

Reviewer #1 (Remarks to the Author):

In the manuscript entitled "The neutrophil antimicrobial peptide cathelicidin promotes Th17 differentiation" Minns et al. investigate the propensity of cathelicidin to drive helper T cell differentiation into a specific Th17 phenotype and suppression of TH1/Th2 differentiation. The authors found that cathelicidin potentiated in vitro differentiation of Th17 through a partially AHR and TGFβ dependent mechanism. Additionally, cathelicidin inhibited Th1 differentiation. The authors noted that cathelicidin promoted Th17, but not Th1, survival, which may explain this discrepancy. Using an in vivo salmonella vaccination model with cathelicidin deficient mice, Minns et al. suggested that cathelicidin derived from lymph node neutrophils is required for T cell IL-17 production. Finally, the authors demonstrate that their findings may have some translational potential. This work has implications for the understanding of the basic mechanisms of T cell differentiation and may ultimately be leveraged for developing treatments for Th17 mediated autoimmune diseases like psoriasis or in infectious diseases in which a Th17 response is desirable.

These are very interesting and important observations. Below are some comments that would improve the level of evidence they provide to support their conclusions.

Major comments:

- a. Apart from IL-6, IL-23, TGFβ1 and TH1/Th2 neutralizing antibodies, IL1B (PMID: 29514093) is known to be an important cytokine for differentiation of CD4 T cells into a Th17 phenotype. Why did the authors omit the use of IL1B in their differentiation cocktail?
- b. It is concerning that the authors only see ~1% IL-17 producing cells following their differentiation protocol with sort purified cells (Figure 1C). I am skeptical as to whether these cells have in fact been differentiated. Can the authors please clarify this?
- c. The suppression of Th1 differentiation under TH17 driving conditions by cathelicidin is indeed interesting. However, this is expected given the reprogramming towards Th17 phenotype. Can the authors do the same experiment under Th1 (IL12, IL2, anti-IL4Ab) and Th2 (IL-4, IL-2, Anti-IFNγ Ab) driving conditions in the presence of cathelicidin and show that there is suppression of Th1 (and also possibly Th2) differentiation. Additionally, it is concerning that up to 80% of the Th17 skewed cells express Tbet, the master Th1 transcriptional regulator. A lower frequency of cells expresses RORγt. Are these in fact Th17s?
- d. The authors should, at all stages of their investigation, show that their differentiated helper T cells are phenotypically Th17s and not Th1s and Th2s. Further, assessment of additional endpoints, such as STAT3 signaling, may support their conclusions.
- e. In figure 8 B, can the authors please include the data on IL-17F and IL-22 production from human CD4 differentiated Th17 cells.
- f. Since the authors show that the effect of cathelicidin is independent of the typical receptor based signaling through FPRL or lipid raft mediated activation, would the authors comment on how cathelicidin acts in conjugation with TGFβ? Does cathelicidin enhance TGFβ receptor and its downstream signaling to activate TH17 cells? Can the authors please clarify this with signaling (phosphor-blots) and loss of receptor (TGFβR) function experiments?
- g. The authors should include representative plots for all markers shown in graphical format (i.e. Tbet, Th2 cytokines, Th1 cytokines, etc.).
- h. In my copy of the manuscript, the images of several of the representative flow plots shown are relatively low quality and difficult to interpret.

Other potential areas for improvement but not essential

- a. The involvement of AHR and requirement of TGFβ1 indicates that cathelicidin is actually driving the pre-activated TH17 differentiation state, rather than by itself activating a TH17 differentiation pathway. Can the authors utilize the TGFβ receptor knockout mouse model and cross it with cathelicidin knockout mice and show that exogenous addition of cathelicidin would show no effect

on TH17 differentiation. What is the down-stream mechanism to AHR activation potentiating TH17 differentiation in the presence of cathelicidin.

b. Determining whether neutrophil-derived cathelicidin drives Th17 responses in a clinically relevant model would improve the significance of this work. Given that this research group focuses on multiple sclerosis, I suggest the addition of an EAE model.

c. Additionally, if the authors have access to blood from psoriasis patients, they could isolate neutrophils from psoriasis patients and show that these neutrophils under degranulating conditions show significant increase in Th17 differentiation in comparison to neutrophils from healthy patients. This would increase the translational aspect of their results.

Minor comments:

1. The authors should look for IL-17F by ELISA as was done for IL-17A and IL-22 to ensure IL-17F is secreted.

2. Figure 1B: Suggest including a key for the different colored symbols.

3. Suggest rephrasing "...strikingly by a far greater amount..." as this is somewhat misleading as written – yes, the fold increase is large, but the magnitude is relatively small.

4. "...dependent on concentration of cathelicidin...and time in culture...". The authors should perform statistical analysis to support this statement.

5. Suggest rephrasing "...neutrophil peptide" when referring to cathelicidin as it is produced by a number of cell types as noted in the introduction. The authors should also include adipocytes to the list of cell types able to produce.

6. Suggest rephrasing section about "normal development of Th17". The developmental process of helper T cells is the same. Cathelicidin appears to play a role in active differentiation rather than previously differentiated cells.

7. Suggest mentioning that the results in Figure 8b were not significant

Reviewer #2 (Remarks to the Author):

This manuscript provides data to indicate that cathelicidin production by neutrophils enhances the differentiation of Th17 cells and induces IL-17F production. Cathelicidin also suppresses Th1 differentiation and protects Th17 cells from apoptosis. Using a mouse model, the authors demonstrate that heat-killed *Salmonella typhimurium* induces cathelicidin production from neutrophils, which contributes to Th17 responses. Cathelicidin deficient mice have reduced Th17 responses in response to heat-killed *Salmonella typhimurium*. Although we are increasingly aware of the diverse functions of neutrophils, this is the first manuscript to clearly define a mechanism whereby neutrophils instruct Th17 responses. Th17 responses then drive neutrophil recruitment and function, providing a feed-forward loop of amplification.

Most of the findings are novel, the manuscript is well-written, and the experimental design/statistical analyses are appropriate. The topic of research is of great interest to immunologists, specifically to anyone interested in inflammation, neutrophils, and T helper subsets. In general, the data are well-presented, although several recommendations are made to improve the clarity. The authors appropriately reference and discuss previous findings related to their data and appropriately indicate where their data confirm, and also expand upon, previous findings. Several comments should be addressed by the authors to improve the overall quality of the manuscript.

Major comments:

1. To make the claim that cathelicidin functions in a receptor independent manner, the authors should use cells from mice lacking P2XR and FPR2, since these are known to be receptors for cathelicidin.

2. The authors show that cathelicidin increases the aryl hydrocarbon receptor and suggest this is one mechanism leading to its ability to enhance Th17 differentiation. However, if there is not increased expression of the ligand(s) for AHR, then how do the authors predict the upregulation of the receptor enhances Th17 differentiation? Are ligands for AHR also upregulated by cathelicidin?

Minor comments:

1. Figure 1A shows a P value, but the statistical analysis used is not described.
2. A figure legend/key needs to be provided for Figure 1B (what are the filled and open circles representing?).
3. In several places in the manuscript, the authors refer to IL-17, while in others they specifically state IL-17A or IL-17F. Presumably the use of IL-17 refers to IL-17A, but this should be consistent throughout the manuscript.
4. In the legend for Figure 6, the authors state "...5M OT-II cells...." Presumably 5M means 5 million, but this should be stated.
5. The data in figure 7C seem to be showing neutrophils producing cathelicidin after isolation of cells from HKST treated mice. However, the figure legend states that "neutrophil numbers" are shown. The results section states that 85% of the cathelicidin signal is associated with Ly6G+ cells. However, it is not clear how that number was determined. Please clarify what is shown in figure 7C and provide actual quantitative data.
6. Provide purity data from cellular isolations and cell sorting experiments for the CD3, CD4, and CD8 T cell subsets.

Reviewer #3 (Remarks to the Author):

Minns et al describe a novel role for neutrophil-produced cathelicidin in promoting Th17 differentiation, revealing a feedback between neutrophils and Th17 cells that intuitively corresponds to the presence of these cells in IL-17-associated inflammation and suggests more of a co-dependence than previously thought. Overall, the manuscript is very well-written and clear, and the experiments as presented are convincing for a role of cathelicidin in Th17 biology and a partial mechanism through AHR induction. However, there are some areas that could be improved, particularly on whether this pathway is important in vivo for function of Th17 cells: the authors show vaccination models but do not test infection clearance, and in some of the more mechanistic experiments as detailed below.

The authors propose that cathelicidin enhances survival of Th17 cells, which would certainly fit with the often chronic nature of Th17-associated autoimmune conditions and is therefore potentially an important observation. However, they miss the opportunity to demonstrate this in vivo in the OTII system: what are the % and numbers of OTII+ cells in the WT and KO mice? Does reduced death in vitro also correspond to increased numbers? Although they show CFSE labeling (as geo mean which is a little unusual), better efforts at demonstrating proliferation is not altered could be undertaken. Is the enhanced survival due to the reduced IL-2 production as shown in Fig 3A? This would correspond to the Fas/FasL changes and suggest an indirect rather than direct mechanism of Cathelicidin in T cell survival.

In Figure 7, adding WT or cathelicidin-/- neutrophils to WT Th17 cultures is a good idea to more directly demonstrate neutrophil-produced cathelicidin role, but why taken from naïve bone marrow? Wouldn't immunized LN/spleen be a more representative source as this could change neutrophil production of inflammatory mediators? The data in D,E is not that convincing as only half the animals respond to neutrophils in the WT group and the UT control is different between WT and KO groups, so some work on providing a more clear demonstration that neutrophils promote Th17 through cathelicidin is needed. For example, could you deplete neutrophils in WT vs KO mice: one would predict a change in Th17 in WT but not KO?

How is cathelicidin detected by T cells, and what is known about signaling mechanisms that allow it to achieve these effects? These should at least be discussed. Is it possible that Th1/Th2 cells do not have the receptors (is this a function of TGFb1?).

Kingston Mills' group has suggested that neutrophils recruited to inflamed LN produce IL-1 to promote Th17 differentiation. Can the authors confirm that neutrophil recruitment and IL-1 production is not deficient in their knockout mice?

We thank the reviewers for their careful and expert reading of our manuscript. Despite significant difficulties in performing lab work under current COVID-related restrictions, we have answered the vast majority of their questions and believe our work to be much stronger as a result. As a result of the changes, we have added an extra figure as well as many extra panels to the existing figures. We have highlighted all the changes in yellow in the updated manuscript.

Yours sincerely

Emily Findlay (corresponding author)

Reviewer #1 (Remarks to the Author):

In the manuscript entitled "The neutrophil antimicrobial peptide cathelicidin promotes Th17 differentiation" Minns et al. investigate the propensity of cathelicidin to drive helper T cell differentiation in to a specific Th17 phenotype and suppression of TH1/Th2 differentiation. The authors found that cathelicidin potentiated in vitro differentiation of Th17 through a partially AHR and TGF β dependent mechanism. Additionally, cathelicidin inhibited Th1 differentiation. The authors noted that cathelicidin promoted Th17, but not Th1, survival, which may explain this discrepancy. Using an in vivo salmonella vaccination model with cathelicidin deficient mice, Minns et al. suggested that cathelicidin derived from lymph node neutrophils is required for T cell IL-17 production. Finally, the authors demonstrate that their findings may have some translational potential. This work has implications for the understanding of the basic mechanisms of T cell differentiation and may ultimately be leveraged for developing treatments for Th17 mediated auto-immune diseases like psoriasis or in infectious diseases in which a Th17 response is desirable.

These are very interesting and important observations. Below are some comments that would improve the level of evidence they provide to support their conclusions.

We thank the reviewer for these kind comments.

Major comments:

a. Apart from IL-6, IL-23, TGF β 1 and TH1/Th2 neutralizing antibodies, IL1 β (PMID: 29514093) is known to be an important cytokine for differentiation of CD4 T cells into a Th17 phenotype. Why did the authors omit the use of IL1 β in their differentiation cocktail?

We agree with the reviewer that IL-1 β is known to be important for the differentiation of Th17 cells and that we should have included this in our original manuscript. We have now performed our Th17 differentiation in the presence of IL-1 β , and we find that cathelicidin also significantly increases Th17 differentiation in the presence of IL1 β – but only in the

presence of TGF- β . That is, cathelicidin does not enhance IL-1 β signalling. We have included this data in Fig.2F and discussed it in the text in page 7, lines 1-3.

b. It is concerning that the authors only see ~1% IL-17 producing cells following their differentiation protocol with sort purified cells (Figure 1C). I am skeptical as to whether these cells have in fact been differentiated. Can the authors please clarify this?

The low production of IL-17A in purified CD4+ T cells (which are also ROR γ t+ Tbet- cells) in our previous Figure 1C (now Figure 1K) was because the timepoint shown was after 48 hours' culture. We have often noted that sort purified CD4+ T cells are slower to differentiate to Th17 cells than whole splenocyte populations. To clarify this point, we have now included the results from 72 hours culture of purified cells. This demonstrates that after 72 hours in culture an average of 11% of untreated CD4+ T cells (in Th17 driving conditions) are ROR γ t+ IL17A+. This is significantly enhanced by cathelicidin. This is representative of other publications (Veldhoen Immunity 2006, Ivanov Cell 2006).

This data has been updated in Fig.1K and in the text at page 6, lines 1-5.

c. The suppression of Th1 differentiation under TH17 driving conditions by cathelicidin is indeed interesting. However, this is expected given the reprogramming towards Th17 phenotype. Can the authors do the same experiment under Th1 (IL12, IL2, anti-IL4Ab) and Th2 (IL-4, IL-2, Anti-IFN γ Ab) driving conditions in the presence of cathelicidin and show that there is suppression of Th1 (and also possibly Th2) differentiation

These experiments have now been done. We show that suppression of Th1 differentiation, as read by measurement of IFN- γ , was only suppressed by cathelicidin in the presence of TGF- β 1, and not under Th1-driving or Th2-driving conditions. We have shown this data in Fig.5I and 5J and discuss it in page 10, lines 18-20.

Additionally, it is concerning that up to 80% of the Th17 skewed cells express Tbet, the master Th1 transcriptional regulator. A lower frequency of cells expresses ROR γ t. Are these in fact Th17s?

We thank the reviewer for this comment. To clarify that these cells are indeed Th17 cells, we have performed transcription factor / cytokine co-staining in our samples, which previously we had not done. We show that almost 100% of the IL-17A producing cells in our cultures are ROR γ t+. Representative plots showing this have been added to figure 1J and to the text in page 5, lines 20-24.

Next we investigated the Tbet / Ror γ t expression more closely. More recent experiments repeat the lower cluster in Figure 5B, which is that approximately 40% of cells in untreated Th17-skewed cultures express Tbet. The majority of these cells also express ROR γ t (Figure 5E). Double-positive Th17 cells have been previously noted during autoimmunity (Brucklacher-Waldert Brain 2009).

Our analysis of these populations under Th17 driving conditions shows the presence of a very small population of Tbet+ RORyt- cells which produce IFN γ and do not produce IL17 – approximately 1-2% of cells. These are Th1 cells and are almost abolished by cathelicidin exposure (Fig.5F).

There are then RORyt+ Tbet+ and RORyt+ Tbet- cells. These populations both make IL-17 (Figures 5G and 5H). Th17 cells are defined by RORyt expression and IL-17 production and so we classify these cells as such.

Cathelicidin exposure decreases the proportion of double positive Tbet+ RORyt+ cells and increases the single RORyt+ proportion. However, cathelicidin increased production of IL-17A from both these cell subsets – as long as RORyt was expressed, cathelicidin increased the IL-17 production (Fig.5H,I).

To clarify this point we have included a number of extra graphs in Figure 5 E-H and discuss this in the text on page 10, lines 1-12.

d. The authors should, at all stages of their investigation, show that their differentiated helper T cells are phenotypically Th17s and not Th1s and Th2s. Further, assessment of additional endpoints, such as STAT3 signaling, may support their conclusions.

To improve our manuscript we have now included more flow cytometry plots. These demonstrate that all the IL-17-producing cells are RORyt+, as discussed above (Fig.1J).

For further confirmation, we examined STAT3 phosphorylation in these cells, as suggested by the reviewer. STAT3 signalling is required for the differentiation of Th17 and not Th1 cells (Zhou Nature Immunology 2007). We show that the geometric mean of STAT3 phosphorylation and the % of CD4+ cells showing phosphorylated STAT3 are significantly increased following cathelicidin exposure. This data is shown in Fig.1L-N and discussed in the text in page 6 lines 6-8.

We believe this data confirms that our induced cells are indeed Th17 cells and we are grateful to the reviewer for suggesting this experiment and improving the clarity of our paper.

e. In figure 8 B, can the authors please include the data on IL-17F and IL-22 production from human CD4 differentiated Th17 cells.

This has now been completed. IL-17F was increased by cathelicidin in human cells (Fig.9D). Interestingly, in human cells IL-22 was suppressed by cathelicidin exposure (Fig.9E). These data have now been discussed in the text on page 15 lines 8-10.

f. Since the authors show that the effect of cathelicidin is independent of the typical receptor based signaling through FPRL or lipid raft mediated activation, would the authors comment on how cathelicidin acts in conjugation with TGF beta? Does cathelicidin enhance TGFbeta receptor and its downstream signaling to activate TH17 cells? Can the

authors please clarify this with signaling (phosphor-blots) and loss of receptor (TGFB β) function experiments?

We agree this is an important point. To answer this point, we firstly examined whether cathelicidin increased IL-23R, TGFB β 1 or IL-6R expression on the CD4⁺T cells. Following exposure to cathelicidin no difference in these receptors was noted on the T cells. This data is now included in Figure 2B and C and discussed in the text on page 6 lines 17-20.

To examine downstream TGF β signalling, we have examined the phosphorylation of Smad2/3, which occurs following TGF- β 1 signalling (Nakao EMBO J 1997). Following 24 hours' exposure of sorted CD4⁺ T cells to cathelicidin, in Th17 driving conditions, SMAD2/3 phosphorylation was significantly enhanced compared to UT Th17 cells (Figure 2D,E).

These data have been discussed in page 6 lines 20-23 and in the discussion on page 16 lines 15-20.

Our current hypothesis is that cathelicidin is able to enhance TGF- β 1 signalling through boosting of SMAD2/3 phosphorylation, leading to enhanced AhR expression. This hypothesis is the basis for ongoing work to follow up this study.

We thank the reviewer for suggesting this experiment, which has led to interesting new findings.

g. The authors should include representative plots for all markers shown in graphical format (i.e. Tbet, Th2 cytokines, Th1 cytokines, etc.).

We have now included plots showing representative staining for all markers assessed by flow cytometry.

h. In my copy of the manuscript, the images of several of the representative flow plots shown are relatively low quality and difficult to interpret.

We apologise for this and have now increased the resolution of all our images. All figures have now been made in Adobe Illustrator to improve their quality.

Other potential areas for improvement but not essential

a. The involvement of AHR and requirement of TGFB β 1 indicates that cathelicidin is actually driving the pre-activated TH17 differentiation state, rather than by itself activating a TH17 differentiation pathway. Can the authors utilize the TGF beta receptor knockout mouse model and cross it with cathelicidin knockout mice and show that exogenous addition of cathelicidin would show no effect on TH17 differentiation. What is the down-stream mechanism to AHR activation potentiating TH17 differentiation in the presence of cathelicidin.

We thank the reviewer for this future experimental suggestions, but this in vivo modelling is not within the scope of this manuscript. We agree that cathelicidin does not by itself promote

Th17 differentiation, but potentiates the differentiation induced already by TGF- β 1, as demonstrated by the data in Fig.1C showing that cathelicidin alone cannot induce Th17 differentiation.

It has been previously shown that TGF- β 1 signalling induces AhR expression (Alves de Lima 2018, Garcia Perez 2020), and we currently hypothesise that cathelicidin enhances this through its boosting of TGF β 1 signalling.

b. Determining whether neutrophil-derived cathelicidin drives Th17 responses in a clinically relevant model would improve the significance of this work. Given that this research group focuses on multiple sclerosis, I suggest the addition of an EAE model.

We agree with the reviewer that an additional model would be useful. We were not able to perform long EAE experiments for this paper under the restrictions imposed on us by COVID-19. However, we have been able to perform an influenza model in wildtype and cathelicidin knockout mice, and examine cytokine production in the draining mediastinal lymph nodes of day 3 post infection. We show that ex vivo (unstimulated) T cells from the lymph nodes of infected wildtype mice produce significantly more IL-17A than T cells from knockout mice (Figure 8J).

We have previously demonstrated that cathelicidin enhances a protective host response to influenza, improving survival to an extent that is not explained by a minor impact on viral load (Barlow, Svoboda et al Plos One 2011), but the mechanisms involved were unclear. These new data support those conclusions in a clinically relevant model, and, we believe, improve the significance of this work, using an additional, very different, in vivo model.

We discuss these results in the text on page 14 line 19- page 15 line 2.

c. Additionally, if the authors have access to blood from psoriasis patients, they could isolate neutrophils from psoriasis patients and show that these neutrophils under degranulating conditions show significant increase in Th17 differentiation in comparison to neutrophils from healthy patients. This would increase the translational aspect of their results.

This is an interesting suggestion, but is currently beyond the scope of this manuscript. To investigate the potential to complete this experiment, we contacted Professor Richard Weller, consultant dermatologist at the University of Edinburgh. Prof. Weller informed us that with online clinics due to COVID-19 and the delays in approval of ethics for new projects, it would be at least next year before such experiments are possible. As a result, the idea will be retained for possible future study.

In addition, it is worth noting that the majority of cathelicidin produced in psoriasis is produced by the keratinocytes, not the neutrophils (Lande Nature 2007, Frohm JBC 1997) and we are unsure whether neutrophils from psoriasis patients would have more cathelicidin,

or release it more quickly, than healthy patients. Careful consideration and validation of the basis for this idea would be required.

Minor comments:

1. The authors should look for IL-17F by ELISA as was done for IL-17A and IL-22 to ensure IL-17F is secreted.

IL-17F ELISAs have now been performed. They are shown in Figure 3E and discussed in the text on page 8 lines 2-4.

2. Figure 1B: Suggest including a key for the different colored symbols.

We apologise and have now included a key for all panels.

3. Suggest rephrasing "...strikingly by a far greater amount..." as this is somewhat misleading as written – yes, the fold increase is large, but the magnitude is relatively small.

This has now been re-worded.

4. "...dependent on concentration of cathelicidin...and time in culture...". The authors should perform statistical analysis to support this statement.

We have performed ANOVA analysis on Figure 4E and F which confirms both length of exposure and concentration of cathelicidin enhance AHR expression. As we are unable to do two-way ANOVA on these data (as the experiments were separate) we have rephrased and toned down this statement (page 8 lines 19-20).

5. Suggest rephrasing "...neutrophil peptide" when referring to cathelicidin as it is produced by a number of cell types as noted in the introduction. The authors should also include adipocytes to the list of cell types able to produce.

These changes have now been made throughout, and adipocyte expression added to page 4 line 4.

6. Suggest rephrasing section about "normal development of Th17". The developmental process of helper T cells is the same. Cathelicidin appears to play a role in active differentiation rather than previously differentiated cells.

We agree with the reviewer, and have re-written all these phrases to clarify our point that cathelicidin potentiates Th17 differentiation during inflammation.

7. Suggest mentioning that the results in Figure 8b were not significant
Having performed experiments with new donors to measure IL17F, the results in 8b (now

Fig.9C) are now significant with the increased donor number, and the figure has been updated.

Reviewer #2 (Remarks to the Author):

This manuscript provides data to indicate that cathelicidin production by neutrophils enhances the differentiation of Th17 cells and induces IL-17F production. Cathelicidin also suppresses Th1 differentiation and protects Th17 cells from apoptosis. Using a mouse model, the authors demonstrate that heat-killed *Salmonella typhimurium* induces cathelicidin production from neutrophils, which contributes to Th17 responses. Cathelicidin deficient mice have reduced Th17 responses in response to heat-killed *Salmonella typhimurium*. Although we are increasingly aware of the diverse functions of neutrophils, this is the first manuscript to clearly define a mechanism whereby neutrophils instruct Th17 responses. Th17 responses then drive neutrophil recruitment and function, providing a feed-forward loop of amplification.

Most of the findings are novel, the manuscript is well-written, and the experimental design/statistical analyses are appropriate. The topic of research is of great interest to immunologists, specifically to anyone interested in inflammation, neutrophils, and T helper subsets. In general, the data are well-presented, although several recommendations are made to improve the clarity. The authors appropriately reference and discuss previous findings related to their data and appropriately indicate where their data confirm, and also expand upon, previous findings. Several comments should be addressed by the authors to improve the overall quality of the manuscript.

Major comments:

1. To make the claim that cathelicidin functions in a receptor independent manner, the authors should use cells from mice lacking P2XR and FPR2, since these are known to be receptors for cathelicidin.

Cathelicidin has multiple receptors, two of the major ones being P2X7R and FPR2. It has also been demonstrated to bind other receptors including EGFR, CXCR2, MrgX2, other unidentified GPCR, and to bind to GAPDH intracellularly (Dorin et al Molecular Medical Microbiology 2014), and also in multiple studies appears to act in a receptor independent manner. Based on our data using the D-enantiomer of LL-37, which would not bind to the cognate receptor for LL-37, we suggested that in this system cathelicidin is acting in a receptor-independent fashion.

We were unable to use the suggested knockout mice in this study owing to time and import restrictions during COVID-19, so to answer this question we used the chemical inhibitors of P2XR and FPR2 most commonly applied in studies that have implicated these receptors in

cathelicidin research; oATP and WRW4 respectively. We found that WRW4, the inhibitor of FPR2, had no effect on cathelicidin's enhancement of Th17 differentiation. However, we show that oATP – an inhibitor of P2X7 – gave results more difficult to interpret, and appeared to suppress cathelicidin-mediated potentiation of Th17 differentiation. oATP was toxic to cells at the higher concentrations used and altered the IL-17A production even in the absence of cathelicidin (Figure 2I), and statistical analysis with a two-way ANOVA shows that cathelicidin and oATP do not interact to enhance IL-17 production. We also must bear in mind that oATP is not only a P2X7 inhibitor but also attenuates inflammation by P2X7-independent mechanisms (Beigi Br J Pharmacol 2003).

We thank the reviewer for suggesting this experiment, which has given rise to interesting data. Future studies in our lab will involve examining cathelicidin's influence on T cell function in knockout mice for P2X7. We have added the new data to Figure 2H and I and we discuss it now in the results (page 7 lines 13-18) and in the discussion (page 18 lines 6-16).

2. The authors show that cathelicidin increases the aryl hydrocarbon receptor and suggest this is one mechanism leading to its ability to enhance Th17 differentiation. However, if there is not increased expression of the ligand(s) for AHR, then how do the authors predict the upregulation of the receptor enhances Th17 differentiation? Are ligands for AHR also upregulated by cathelicidin?

This is an excellent point which we thank the reviewer for raising, and which we should have discussed more in the original manuscript.

Our hypothesis has been that cathelicidin enhances expression of AhR and that this allows enhanced signalling by the ligands available in the cell culture medium, by increasing sensitivity to available ligands. To examine this, we used IMDM medium in contrast to RPMI, which we had used previously. IMDM contains more AhR ligands relating from tryptophan metabolism than RPMI and their excess results in increased Th17 differentiation (Veldhoen J Exp Med 2009). We show that cathelicidin enhances Th17 differentiation equally in both media. This experiment demonstrates that increasing AhR ligands can increase Th17 differentiation, but that in both high and lower ligand levels we can enhance this differentiation through cathelicidin's boosting of the AhR.

These data have been added to figure 4I and are discussed in the text in page 9 lines 9-19.

Minor comments:

1. Figure 1A shows a P value, but the statistical analysis used is not described.

This test (a two-tailed t test on paired raw data before conversion to %) has now been added to the figure legend.

2. A figure legend/key needs to be provided for Figure 1B (what are the filled and open circles representing?).

This has now been added to the figure (which is now figure 1C).

3. In several places in the manuscript, the authors refer to IL-17, while in others they specifically state IL-17A or IL-17F. Presumably the use of IL-17 refers to IL-17A, but this should be consistent throughout the manuscript.

These have now been noted and corrected to be more specific.

4. In the legend for Figure 6, the authors state "...5M OT-II cells..." Presumably 5M means 5 million, but this should be stated.

This has now been corrected.

5. The data in figure 7C seem to be showing neutrophils producing cathelicidin after isolation of cells from HKST treated mice. However, the figure legend states that "neutrophil numbers" are shown. The results section states that 85% of the cathelicidin signal is associated with Ly6G+ cells. However, it is not clear how that number was determined. Please clarify what is shown in figure 7C and provide actual quantitative data.

We apologise for this. This image (which is now Fig.8C) shows a representative image of cathelicidin-Ly6G co-staining in the lymph node. We counted the total number of cathelicidin+ cells in the lymph nodes and calculated the percentage of these cells which also showed Ly6G staining, indicating that they are neutrophils. Other cell types produce cathelicidin and so our aim in this panel was to demonstrate that the majority of cathelicidin-producing cells in the lymph nodes were neutrophils. We have clarified this in the text and added quantitative data on page 13 line 21 – page 14 line 1.

6. Provide purity data from cellular isolations and cell sorting experiments for the CD3, CD4, and CD8 T cell subsets.

The purity achieved by CD4+ cell isolations (always over 95%) has now been added to the methods in page 21 lines 13-16. We apologise for the line stating CD3 or CD8 T cell isolations were performed which was previously in the methods – this was mistakenly included in the original manuscript and has now been removed. We have also added a gating strategy for identifying CD4+ T cells from splenocyte populations (Supplementary Figure 1).

Reviewer #3 (Remarks to the Author):

Minns et al describe a novel role for neutrophil-produced cathelicidin in promoting Th17

differentiation, revealing a feedback between neutrophils and Th17 cells that intuitively corresponds to the presence of these cells in IL-17-associated inflammation and suggests more of a co-dependence than previously thought. Overall, the manuscript is very well-written and clear, and the experiments as presented are convincing for a role of cathelicidin in Th17 biology and a partial mechanism through AHR induction. However, there are some areas that could be improved, particularly on whether this pathway is important in vivo for function of Th17 cells: the authors show vaccination models but do not test infection clearance, and in some of the more mechanistic experiments as detailed below.

We thank the reviewer for their comments. We have now added analysis of IL-17 during an infection model to Figure 8J, showing cathelicidin KO mice produce less IL-17A in the lymph node during PR8 influenza infection.

The authors propose that cathelicidin enhances survival of Th17 cells, which would certainly fit with the often chronic nature of Th17-associated autoimmune conditions and is therefore potentially an important observation. However, they miss the opportunity to demonstrate this in vivo in the OTII system: what are the % and numbers of OTII+ cells in the WT and KO mice? Does reduced death in vitro also correspond to increased numbers?

We thank the reviewer for this comment and agree this could be an important mechanism in chronic autoimmunity.

We re-analysed the data from our OT-II experiment to answer this question. We find that indeed the % and number of CD45.1 OT-II cells in the lymph node of KO mice is suppressed compared to WT mice in the lymph node and spleen. This supports our in vitro findings and we are grateful to the reviewer for suggesting this analysis. This data is now in Figure 7F-G and discussed in page 13 lines 1-3.

Although they show CFSE labeling (as geo mean which is a little unusual), better efforts at demonstrating proliferation is not altered could be undertaken. Is the enhanced survival due to the reduced IL-2 production as shown in Fig 3A? This would correspond to the Fas/FasL changes and suggest an indirect rather than direct mechanism of Cathelicidin in T cell survival.

We agree with the reviewer that our proliferation analyses were not optimal in the original manuscript. To improve this analysis we set up our Th17 cultures as before but left them for 72 hours rather than 48, which allowed clear observation of peaks in the CFSE dilution experiment. We found, to our surprise, that cathelicidin suppressed proliferation of RORγt+ cells significantly following 72 hours in culture, which was not evident at 48 hours (Fig.6H-J). The new flow cytometry plots are shown in Fig.6H. This agrees with previous data showing neutrophils can suppress T cell proliferation (Pillay JCI 2012, Aarts Front Immunol 2019) and adds a further mechanism by which cathelicidin alters T cell behaviour.

To test whether the enhanced survival and proliferation noted was related to IL-2 concentration, we spiked the cultures with recombinant IL-2 at concentrations up to 50IU/ml. The CD4+ T cell survival was still enhanced by cathelicidin with IL-2 added into these cultures, suggesting that these two phenomena are unrelated. Likewise, proliferation was unaffected by spiking of IL-2 into the cultures over 72 hours. These data have been added to Figure 6 and discussed in the text at page 11 line 16 – page 12 line 2.

These data are very interesting and we thank the reviewer for suggesting these experiments. We are unclear as to precisely how cathelicidin is suppressing proliferation of these cell subsets and whether the mechanism is direct or indirect, and this is the focus of future experiments in the lab.

In Figure 7, adding WT or cathelicidin-/- neutrophils to WT Th17 cultures is a good idea to more directly demonstrate neutrophil-produced cathelicidin role, but why taken from naïve bone marrow? Wouldn't immunized LN/spleen be a more representative source as this could change neutrophil production of inflammatory mediators?

Unfortunately, we are not able to isolate enough neutrophils from immunised LN or spleen to perform these experiments. To solve this problem, in all of our neutrophil experiments, neutrophils are treated with cytochalasin B and fMLf to prime the cells and promote degranulation in culture (Honeycutt and Niedel J Biol Chem 1986, Sato Mol Cell Biol 2013, Mitchell AJPCP 2008). As neutrophils do not transcribe a great deal in the periphery we believe primed bone marrow neutrophils are a good model of LN cells.

In addition, the production of cathelicidin is unchanged in the periphery to our knowledge. It is produced very early in development, at the myelocyte and metamyelocyte stages and pre-stored in granules (Sorensen Blood 1997), meaning that mature neutrophils from the bone marrow have equivalent stores to those in the LN or spleen.

The data in D,E is not that convincing as only half the animals respond to neutrophils in the WT group and the UT control is different between WT and KO groups, so some work on providing a more clear demonstration that neutrophils promote Th17 through cathelicidin is needed. For example, could you deplete neutrophils in WT vs KO mice: one would predict a change in Th17 in WT but not KO?

We agree with the reviewer that this is an important experiment.

Depleting neutrophils in vivo is difficult as it has been demonstrated to lead to an increase in IL-17 and IL-23 production (Patel Sci Imm 2019). We have ourselves observed a large increase in IL-17A production in WT mice following neutrophil depletion – perhaps owing to widespread neutrophil death - and as such believe this is not the best model to use for our experiments.

In the initial experiments shown all of the cultures did increase IL-17A production in response to neutrophil co-culture, although some by a smaller amount (with the minimum increase

being from 13.1% in untreated wells to 14.5% with neutrophils). We have added extra cultures now to this experiment (Figure 8F) which increases the statistical significance.

To answer this question more completely, we performed an additional experiment in which we co-cultured isolated CD4⁺ T cells with primed WT neutrophils in the presence of a blocking antibody to mouse cathelicidin. In this experiment neutrophils enhanced IL-17A production from the T cells and this was inhibited by the presence of the anti-cathelicidin antibody. This has now been added to Figure 8I and discussed in the text at page 14 lines 18-19.

Finally, we have also analysed the AhR expression following co-culture of T cells with WT and KO neutrophils, as this was missing from the original manuscript. We show that AhR is significantly lower in those T cells co-cultured with KO neutrophils compared to those co-cultured with WT neutrophils (Fig.8, page 14 line 16-17).

How is cathelicidin detected by T cells, and what is known about signaling mechanisms that allow it to achieve these effects? These should at least be discussed. Is it possible that Th1/Th2 cells do not have the receptors (is this a function of TGFb1?).

We agree this is a fundamental point which we should have discussed in more detail in the original manuscript. Our new Figure 2 which we made in response to reviewer 1 (please see comments to that reviewer) demonstrates that cathelicidin can act in a receptor-independent fashion but the Th17 potentiation may also be partially reliant on P2X7 receptor signalling. Cathelicidin can move into other cell types – including plasmacytoid DC and fibroblasts – without a receptor, via endocytosis. We hypothesise that this is also occurring in T cells, and our data showing that the D-enantiomer can potentiate Th17 signalling supports this conclusion. However, as discussed in the response to reviewer 1, this is an important area on which significant future work in our lab will focus.

We agree that the signalling analysis was sparse in the original manuscript. We have now added significant data showing that cathelicidin enhances TGF- β 1 signalling through enhanced phosphorylation of Smad2/3 and STAT3 (Figures 1L-N, 2D-E). This is discussed in the text on page 6 lines 6-8, page 6 lines 20-23 and page 16 lines 16-21.

Kingston Mills' group has suggested that neutrophils recruited to inflamed LN produce IL-1 to promote Th17 differentiation. Can the authors confirm that neutrophil recruitment and IL-1 production is not deficient in their knockout mice?

This is an excellent point and one we should have addressed in the original manuscript. To answer this we have immunised WT and cathelicidin KO mice with heat-killed salmonella and assessed draining popliteal lymph nodes 24 hours later, as this is the peak time for neutrophil infiltration. Firstly, we noted that neutrophil recruitment is unaltered in the KO mice (Figure 8D). Secondly, we detected IL-1 β production by ELISA and found this not to be significantly altered in KO mice compared to WT (Fig 8E). We discuss these data on page 14 lines 3-7.

REVIEWERS' COMMENTS

Reviewer #1 (Remarks to the Author):

I have only minor additional suggestions

Figure 7A/B: label groups

Figure 2I: Show stats between UT and Cath

Figure 4B/C: Capitalize first letter in gene names

Reviewer #2 (Remarks to the Author):

The authors have responded appropriately to my comments and concerns.

Reviewer #3 (Remarks to the Author):

I thank the authors for thoroughly addressing my concerns, and for the clear explanations to my questions.

I don't have any more concerns with this manuscript, which provides an interesting new angle to the Th17:neutrophil interaction in inflammation.

Reviewer requests

Reviewer #1 (Remarks to the Author):

I have only minor additional suggestions

Figure 7A/B: label groups

Figure 2I: Show stats between UT and Cath

Figure 4B/C: Capitalize first letter in gene names

All of these corrections have been made.

Reviewer #2 (Remarks to the Author):

The authors have responded appropriately to my comments and concerns.

Reviewer #3 (Remarks to the Author):

I thank the authors for thoroughly addressing my concerns, and for the clear explanations to my questions. I don't have any more concerns with this manuscript, which provides an interesting new angle to the Th17:neutrophil interaction in inflammation.